# Review of interactive open access publishing with community-based open peer review for improved scientific discourse and quality assurance

Barbara Ervens<sup>1</sup>, Ken S. Carslaw<sup>2</sup>, Thomas Koop<sup>3</sup>, and Ulrich Pöschl<sup>4</sup>

Correspondence: Barbara Ervens (barbara.ervens@uca.fr), Ulrich Pöschl (u.poschl@mpic.de)

Abstract. Scientific discourse and quality assurance can be improved by open access (OA) publishing with public peer review and community discussion. Over 25 years, the viability of this approach has been proven by the interactive OA journal Atmospheric Chemistry and Physics (ACP) and 18 other journals published by the European Geosciences Union (EGU) and its scientific service provider Copernicus Publications. The success of the EGU journals reflects the benefits of community-driven, interactive OA publishing, including high scientific quality and impact, efficient self-regulation, low cost and financial sustainability. Since 2001, EGU has published over 50000 journal articles, 60000 preprints, and 250000 comments, utilizing and integrating different OA financing models (green, gold, diamond/platinum). The EGU journals with multi-stage open peer review are linked to the OA repository and interactive community platform EGUsphere and to the virtual scientific highlight magazine EGU Letters, integrating different levels of scientific communication and exchange. The EGU publications combine multiple features of open science, including different forms of open peer review and community evaluation with open access, open data, and open source elements tailored to the needs and preferences of different disciplines. Indeed, the EGU pioneering approach to transparent peer review has spread to other leading publishers, including the Nature publishing group. We review the approach, achievements and future perspectives of interactive OA publishing (including transformative/institutional agreements and AI/ML tools) and its contribution to a universal epistemic web that captures the scientific discourse and comprehensively documents what we know, how well we know it, and where the limitations are.

#### 1 Introduction

Traditional forms of scientific publishing and peer review do not satisfy current demands for efficient and traceable scientific communication and quality assurance (Pöschl, 2004, 2012; Kriegeskorte, 2012; Bornmann and Haunschild, 2015; Tennant et al., 2017; Ross-Hellauer, 2017; Tennant, 2018; Waltman et al., 2023). To improve the publishing and review process, scientists worked with the European Geosciences Union (EGU) and its publisher Copernicus to develop and launch the first interactive open access (OA) journal *Atmospheric Chemistry and Physics (ACP)* in the years 2000/2001, i.e., a couple of years before the term 'open access' was formally established in the declarations of the *Budapest Open Access Initiative* (2002),

<sup>&</sup>lt;sup>1</sup>Institute of Chemistry, University Clermont Auvergne, CNRS, 63000 Clermont-Ferrand, France

<sup>&</sup>lt;sup>2</sup>Institute for Climate and Atmospheric Science, School of Earth and Environment, University of Leeds, Leeds, LS2 9JT, UK

<sup>&</sup>lt;sup>3</sup>Faculty of Chemistry, Bielefeld University, 33615 Bielefeld, Germany

<sup>&</sup>lt;sup>4</sup>Multiphase Chemistry Department, Max Planck Institute for Chemistry, 55128 Mainz, Germany

Bethesda Statement on Open Access Publishing (2003) and Berlin Declaration on Open Access to Knowledge in the Sciences and Humanities (2003). By now, the EGU publishing portfolio comprises 19 OA journals (Table A1 and EGU journals website) covering the full spectrum of geoscientific research. These interactive journals provide OA to the published articles and allow for public peer review and discussion that is open to the scientific community and to the public. The journals employ public peer review of manuscripts that are posted as preprints or discussion papers after a rapid pre-screening or access review. The advantages of the interactive OA publishing model of ACP and the other EGU/Copernicus journals can be summarized as follows (Pöschl, 2012):

- Free scientific speech and rapid distribution of original research results after quick pre-screening/access review to remove submissions that are clearly deficient or out of scope.
  - Documentation of critical scientific discourse and exchange of arguments, complementary information, open questions,
     scientific controversies or flaws.
  - Traceability of quality assurance by citable reference and permanent digital object identifier (DOI) assigned to all elements of public review and discussion.




- Transparency in maintaining scientific integrity by facilitating the detection and reducing the risk of unethical behavior or abuse of the publication and review process (plagiarism, delay/obstruction during hidden peer review etc.).
- Public exposure, review and discussion of original manuscripts to provide public recognition that attracts high-quality submissions and a permanent record of critical feedback, which, in turn, deters low-quality submissions and ultimately results in low rejection rates.
- Public scrutiny to achieve effective self-regulation, high efficiency of scientific quality assurance and efficient use of (peer) reviewing capacities as the most limited resource in the scientific publishing process.
- Educational value of public access to scientific communication and discussion, enabling everyone to follow and learn from real examples of how scientific critiques are addressed and how consensus can be reached, or how disagreement can be handled in a rational and constructive way.

The motivation, approach and design of the EGU interactive OA publishing model have been described before (Gura, 2002; Pöschl, 2004, 2012; Cartlidge, 2007; Pöschl and Koop, 2008; van Edig, 2016); its performance and benefits have been independently evaluated and compared to other publishing models (Bornmann et al., 2010, 2011, 2012; Ho et al., 2013; Bornmann and Haunschild, 2015; Kovanis et al., 2017; Tennant et al., 2017; Ross-Hellauer and Görögh, 2019; Hynninen, 2022). For example, Bornmann et al. (2011) stated: "All in all, our results on the predictive validity of the ACP peer review system can support the high expectations that Pöschl (2010), chief executive editor of ACP, has of the new selection process at the journal: 'The two-stage publication process stimulates scientists to prove their competence via individual high-quality papers and their discussion, rather than just by pushing as many papers as possible through journals with closed peer review and no direct public feedback and recognition for their work. Authors have a much stronger incentive to maximize the quality of their manuscripts prior to submission for peer review and publication, since experimental weaknesses, erroneous interpretations, and relevant but unreferenced earlier studies are more likely to be detected and pointed out in the course of interactive peer review and discus-

sion open to the public and all colleagues with related research interests. Moreover, the transparent review process prevents authors from abusing the peer review process by delegating some of their own tasks and responsibilities to the referees during review and revision behind the scenes.'

Illustrating the big picture of interactive OA publishing, Figure 1 outlines how public review and discussion contribute to the advancement and refinement ("distillation") of scholarly knowledge and to the development of an epistemic web that displays the scholarly discourse and shows what we know, how well we know it, and where the limitations are in accordance with the scientific method and critical rationalism (Popper, 1974; Hyman and Renn, 2012; Pöschl, 2012; MPIC Open access).


**Figure 1.** Schematic illustration of interactive open access publishing within an epistemic web and global commons of scholarly knowledge (Popper, 1974; Hyman and Renn, 2012; Pöschl, 2012; MPIC Open access)

In Section 2, we provide a brief overview of approaches and developments that evolved over the past decades to enhance the accessibility, efficiency and transparency of the scientific publishing and review process. In Sections 3 and 4, we review the key features and evolution of the interactive OA publishing approach as applied in the *EGU journals* and its related platforms including the OA repository *EGUsphere*, the *Encyclopedia of Geosciences* and the virtual highlight magazine *EGU Letters*.

# 2 Recent developments in scientific publishing

# 2.1 Open access (OA) publishing

#### 2.1.1 Early initiatives on OA publishing

Open access, i.e., the free online availability and re-usability of scientific publications, leads to greater visibility, impact and equitable accessibility of scientific research results and knowledge for the global scientific community and interested public (Berlin Declaration on Open Access to Knowledge in the Sciences and Humanities, 2003; Eysenbach, 2006; Norris et al., 2008; SPARC Europe, 2015; Tennant et al., 2016; Langham-Putrow et al., 2021; Brainard, 2024a; Can et al., 2024; Chen et al., 2024; Huang et al., 2024). In the traditional subscription-based model of scientific publishing, most articles require payment for access, which is particularly disadvantageous for scientific exchange and for people in resource-poor institutions and regions. Early attempts to make scientific journal articles and preprints openly available on the internet began with bottom-up initiatives of scientific researchers in the 1980s and 1990s, including but not limited to the high energy physics preprint repository arXiv (Section 2.3.1), the New Journal of Physics (OA since 1998, Bodenschatz (2008)) and the Journal of Medical Internet Research (OA since 1999, Eysenbach (2019)). In the early 2000s, further OA journals were established by researchers, learned societies and innovative commercial publishers in the geosciences and life sciences, including Copernicus Publications, the European Geosciences Union (EGU) and its predecessor European Geophysical Society (1971 - 2003), the Public Library of Science (PLOS) and BioMed Central. In parallel, major scientific institutions adopted the aims and further developed the concepts of OA publishing. For example, the Berlin Declaration on Open Access to Knowledge in the Sciences and Humanities (2003) has been signed by over 800 leading scholarly organizations worldwide, whereby EGU was one of the first international learned societies among the signatories (EGU News, 2020).

The proportion of OA articles among the large number of scientific journal articles (> 2 million per year, published in > 20,000 peer-reviewed journals, e.g., *Web of Science; Scopus; Ulrich's Web*), however, increased only slowly during the first 10 years after the *Berlin Declaration*, because most funds available to cover costs of scientific journal publishing were bound in subscription contracts. Traditional publishers moved only very slowly and reluctantly towards a proper OA publishing market (Schimmer et al., 2015; Brainard, 2023; Frank et al., 2023; Kiley, 2023). To accelerate the progress and transform the corpus of traditional pay-walled subscription journals to OA, scholars, scholarly organizations and research funders developed a variety of initiatives at institutional, national and international levels (OA2020, 2016a; Pöschl, 2020; cOAlition S Blog, 2023). These and related initiatives aim at redirecting funding from subscription or paywall access to OA in scholarly oriented and cost-efficient, i.e., cost-neutral or cost-saving ways (Pöschl, 2015; Schimmer et al., 2015).

#### 2.1.2 OA models: green, gold, diamond/platinum

Different approaches are employed to provide free access to scientific content. These approaches vary in terms of timing of open access, copyrights, coverage of publications costs and funding sources. The three major forms are often described by the following categories, which are, however, only loosely defined:

- Green OA (OA archiving) requires that authors self-archive their publications in repositories that are funded by their institution or by other sources. Publishers of the journals where the original article was published may impose embargo periods prior to archiving of closed-access articles, and they also may retain the copyrights for the distribution of the work. The green OA approach can help to advance OA, but it has not been proven to offer a viable alternative to traditional journal publishing and quality assurance. It can introduce ambiguities and confusion about the public availability and validity of different versions of scientific studies, where preliminary versions have been self-archived and the final validated version may be available through a subscription journal only. The open science community platform EGU-sphere (Section 4.2) enables the archiving of preprints as well as their linking and transfer to peer-reviewed scientific journals, including but not limited to the EGU interactive OA journals.

- Gold OA (OA publishing) requires the payment of article processing charges (APCs) that are either covered by the authors themselves or via publishing agreements with their institutions. The APCs cover the costs associated with the paper production, archiving and other publisher-related expenses (Section 4.1.7). Gold OA allows immediate and free access to the publication for all readers and, thus, removes the pay-walls that exist in many traditional journals that only allow access when readers or their institutions pay a subscription to the journal. Many current efforts are dedicated to convert such 'closed-access' journals into OA journals via 'transformative agreements' (Section 2.1.3) or to initiate full OA journals, such as those published by EGU/Copernicus. The long-term viability of this approach has been proven for more than a couple of decades by the journals of EGU/Copernicus and other early OA publishers as outlined above and detailed below.
  - **Diamond/platinum OA (OA publishing)** implies that the APCs are covered through funding by institutions, universities, research organizations or other external sources. Even though authors may perceive diamond OA as cost-free publishing, it relies on sustainable funding sources to support and maintain the publishing infrastructure without any charges for authors or readers. Diamond OA is offered by EGU journals during their start-up phase and for corresponding authors from countries in the *Research4Life* groups as well as other authors with insufficient funding (Section 4.1.7). During the past few years, some other, newly launched geoscience journals pursue the diamond OA approach<sup>1</sup>. They are usually sponsored by individual institutions and led by an editorial team that performs all steps of the publishing process for relatively few papers without professional publisher support. Another example is the journal *Aerosol Research*, launched by Copernicus Publications in 2023. This journal is sponsored by several research institutions and scientific societies to allow for long-term financing (Elm et al., 2023). Similar schemes can be envisioned for EGU journals in the future. Depending on scientific developments, disciplinary preferences and the global publishing landscape, sponsors may include research agencies or a sufficient number of large OA publishing agreements. For diamond/platinum OA journals, most of which are relatively recent and small, the large-scale viability, long-term commitment of supporting/funding bodies and scientific sustainability still remain to be proven.

<sup>&</sup>lt;sup>1</sup>Examples include Volcanica (2018), funded by Presses universitaires de Strasbourg, Seismica (2022), funded by McGill Library, Sedimentologika (2023), funded by Bibliothèque de l'Université de Genève and the Society for Sedimentary Geology, Tektonika (2023), funded by University of Aberdeen, Geomorphica (2024), Geodynamica (2025).

#### 2.1.3 Transformative and full OA publishing agreements

Transformative agreements (TAs) or 'publish-and-read' are contracts between scholarly institutions or consortia and publishers to transition publication output to OA; at the same time, these contracts ensure access to content that was previously covered via the institutional subscriptions to the journal. These agreements, thus, lead to OA publication for participating institutions while maintaining access to subscription-based content from other institutions or consortia that have not established OA. Such agreements offer flexible ways for publishers, institutions, consortia and countries of gradually converting subscription-based journals into OA (OA2020, 2016a; ESAC, 2014; cOAlition S Blog, 2023; Springer Nature, 2024). At the same time and as outlined in the OA2020's mission statement (OA2020, 2016b), it remains important to uphold and promote further improvements in scientific publishing at reasonable and competitive costs through innovation and competition. This involves not only traditional subscription publishers through TAs, but also innovative fully OA publishers through equivalent OA publishing agreements (e.g., institutional agreements with Copernicus Publications, Section 4.1.7).

The evolution of the number of TAs in the ESAC registry (2024) that start per year and the number of OA articles published on the basis of TAs are shown in Figure 2. Some examples of such TAs with traditional subscription publishers and of agreements with full OA publishers are listed in Table 1. Leading scholarly organizations and consortia in a growing number of countries have successfully negotiated and used TAs with major international scientific publishers to achieve high percentages of OA for their publication output during recent years (Figure S1 in the Supplement), while some national organizations continue to resist these developments due to concerns over supposed financial sustainability, perceived inequities in cost distribution, or speculative fears about future cost increases (Schuhl, 2024).

Several countries and scholarly organizations have already achieved OA for more than 90% of their scientific journal publishing output (B16 Conference, 2023), using TAs with traditional publishers, agreements with proper OA publishers and further OA publishing funds for scientists to flexibly cover OA article processing charges. For example, the German Max Planck Digital Library (MPDL) has converted more than 95% of the publication output of the Max Planck Society into OA via the German National Consortium DEAL with big traditional publishers (Elsevier, Springer, Wiley) and via individual contracts with numerous OA and learned society publishers (Dér, 2023).

This development towards a full OA landscape helps to maintain and increase bibliodiversity of scholarly scientific publications, i.e., the variety of publishing formats, models, platforms and outlets used to disseminate scientific knowledge in different disciplines, languages and regions of the world (Jussieu Call, 2016). It also includes the range of financing models and mechanisms (e.g., gold/diamond OA) to make scientific publishing and its access inclusive and equitable. This desired outcome will be reached if OA agreements do not only focus on large publishers but are extended to smaller publishers and new publishing models. Acknowledging this evolution toward full OA to the entire scientific literature, the TIME magazine recognized the advances in OA achieved by transformative agreements as one of the '13 Ways the World Got Better in 2023' (TIME, 2023).

The average fee per article in the traditional subscription publishing model is approximately 4000 €, whereas OA journals of similar quality are sustainable with article charges of 2000 € or less (Open APC, 2024). This was shown by Schimmer et al. (2015) in a comprehensive analysis of the global scholarly journal publication market with a financial annual volume

**Figure 2.** a) Number of transformational agreements started per year (left axis) and cumulative number (right axis) registered in the given year with the Initiative for Efficiency and Standards for Article Charges (ESAC, 2014); b) Cumulative number of published open access articles using transformational agreements (figure adapted from Dér (2023) with modifications). The data in the figure are arranged in alphabetical order by country, starting from the bottom.

of ~10 billion €, mostly paid by publicly funded academic libraries. Similar differences were found in more recent, less comprehensive studies (Björk and Solomon, 2015; Pinfield et al., 2016; Triggle and Triggle, 2017; Ross-Hellauer et al., 2018; Pollock and Michael, 2019; Grossmann and Brembs, 2021; Borrego, 2023; Haustein et al., 2024). In other words, the costs of traditional subscription journals are on average higher by a factor of 2 than required to produce high-quality publications. This is also reflected by the high profit margins of traditional journal publishers, often exceeding 35%, i.e., higher oligopoly revenues than in many other industries (Van Noorden, 2013; Larivière et al., 2015; Pöschl, 2020; Butler et al., 2023). Thus, concerns that OA threatens the financial viability of the academic publishing system, as e.g., put forward by Velterop (2003), are unfounded because much more public funding than needed for OA publishing is bound in the traditional subscription journal business. The financial benefits and savings through transformative agreements have been clearly demonstrated by the MPDL and the German DEAL consortium. For example, the MPDL managed to convert practically all scientific journal publication output

**Table 1.** Overview of selected transformative or full open access publishing agreements in various countries (ESAC, 2014; OA2020, 2016a; B16 Conference, 2023)

| Country     | Link to website or document <sup>1</sup>                      | Comment                                                   |
|-------------|---------------------------------------------------------------|-----------------------------------------------------------|
| Austria     | https://at2oa.at/en/                                          | Springer, Wiley, Elsevier                                 |
| Denmark     | https://pro.kb.dk/en/licensing/oa-publication-guidelines/     |                                                           |
|             | guidelines-open-access-publications                           |                                                           |
| Finland     | https://finelib.fi/negotiations/using-oa/                     | Completed or ongoing negotiations                         |
| France      | https://www.couperin.org/category/negociations/               |                                                           |
|             | accords-specifiques-so/                                       |                                                           |
| Germany     | https://deal-konsortium.de/en/agreements                      | DEAL/MPDLS agreements with Wiley, Springer Nature,        |
|             |                                                               | Elsevier                                                  |
|             | https://www.mpdl.mpg.de/21-specials/                          | MPG/MPDL agreements with multiple publishers              |
|             | 50-open-access-publishing.html                                |                                                           |
| Netherlands | https://www.openaccess.nl/en/publishing/publisher-deals       |                                                           |
| Norway      | https://www.openscience.no/en/publisering/apen-publisering    | Full or partial APC coverage, depending on agreement      |
| Sweden      | https://www.kb.se/samverkan-och-utveckling/                   | hybrid and full OA publishers, including PLOS, Copernicus |
|             | oppen-tillgang-och-bibsamkonsortiet/                          |                                                           |
|             | open-access-and-bibsam-consortium/bibsam-consortium/          |                                                           |
|             | open-access-in-bibsam-agreements.html                         |                                                           |
| Switzerland | https://consortium.ch/vertraege-konditionen/?lang=en          | Does not cover articles in special issues                 |
| UK          | https://beta.sherpa.ac.uk/tj-list                             |                                                           |
| USA         | https://osc.universityofcalifornia.edu/for-authors/           | University of California agreement with Elsevier          |
|             | publishing-discounts/elsevier-oa-agreement/                   |                                                           |
|             | https://btaa.org/library/open-scholarship/agreements/         | BTAA <sup>2</sup> agreement with Wiley                    |
|             | wiley-open-access-agreement                                   |                                                           |
|             | https://www.cambridge.org/core/services/open-access-policies/ | BBTA <sup>2</sup> agreement with Cambridge Univ Press     |
|             | read-and-publish-agreements/oa-agreement-btaa                 |                                                           |

<sup>&</sup>lt;sup>1</sup> Last access for all links: 24 Jan 2025, <sup>2</sup> Big Ten Academic Alliance, https://btaa.org/about

of the Max Planck Society from subscription to OA over a period of six years without any increase of expenses in spite of substantial inflation (2018 - 2024: approx 14 million Euros unchanged; see slide 13 in (Dér, 2024); (Dér, 2025). Moreover, the DEAL consortium managed to obtain OA for all research articles with corresponding authors from German institutions while reducing the overall expenses for Elsevier journals from approx 50 million Euros in 2015 to approx 30 million Euros in 2023, corresponding to a 40% cost reduction (Vogel, 2023). The opportunities and viability of cost saving through OA are confirmed by the following: While nearly 50% of new peer-reviewed articles are now published OA, 80% of publisher revenues are still bound in opaque subscription fees (B17 Conference, 2025). On average, 1–2% of total research budgets by scientific

institutions is spent on literature and information provision and publishing (German Science and Humanities Council, 2022). However, the distribution of these funds are important in the OA transition and also implies that financial flows within the institutions should be adjusted accordingly.

OA publishing models have led to valid concerns about the potential impacts of OA on the quality of scientific content (Pöschl, 2004; Björk, 2019; MacLeavy et al., 2020; Frank et al., 2023). Since APCs are levied for individual articles, unlike in subscription payments which are remunerated to entire journals or journal families, every published OA article enhances the publisher profits. As a consequence, the APC business model led to the launch of a large number of journals motivated purely by commercial interests. These journals solicit a high number of submissions that often undergo only cursory (if any) peer review, resulting in publications of low scientific quality and, thus, are frequently referred to as "predatory journals" (Beall, 2017; Grudniewicz et al., 2019; Brainard, 2023). We strongly propose that such economically-driven aberrations of OA publishing have to be counteracted by measures of quality assurance providing transparent evidence of rigorous peer review. With the goal of maintaining or even improving scholarly quality assurance, the EGU interactive OA journals and other innovative OA publishing platforms applying various forms of open and/or transparent peer review and different OA approaches (diamond/gold) were launched even prior to initiatives aimed at the OA transformation of traditional subscription journals (Sections 3 and 4).

# 2.2 Open peer review

Balancing the needs of rapid dissemination of scientific results and publications while ensuring scientific quality can be achieved by complementing traditional publishing practices with open and transparent approaches (Pöschl, 2004, 2010, 2012; Kriegeskorte, 2012; Bornmann and Haunschild, 2015). The traditional peer review system neither allows for tracking the rigor of the peer review, nor does it lead to quick, efficient communication of scientific knowledge (Tennant et al., 2017; Tennant, 2018; Ross-Hellauer, 2017; Borrego et al., 2021; Aczel et al., 2025; Pattinson and Currie, 2025), which may inherently bias scientific assessment and progress (Lee et al., 2013). These shortcomings were recognized in the recent 'Proposal towards responsible publishing' (cOAlitionS, 2023) that suggests measures to ensure scientific quality, several of which are already implemented in the publication model by EGU/Copernicus since 2001 (Section 3).

The need for transparency and the advantages of interactive public discussion during the peer review process have been discussed (Kriegeskorte, 2012; Sandewall, 2012; Walker and Rocha da Silva, 2015; Fiala and Diamandis, 2017; Horbach and Halffman, 2018; Wolfram et al., 2020). During the last decades, new ways of peer review have been suggested, including several forms of open peer review (Ross-Hellauer, 2017). The two most common ones are the disclosure of the reviewer identities to the authors and the publication of reviewer reports alongside the papers either during the peer review or after publication (or a combination of both) (Ross-Hellauer et al., 2023). Initial concerns about potential bias among reviewers in an open peer review process were shown to be unfounded (Thelwall et al., 2020); in fact, there is evidence that such reviews may be even more constructive (Ross-Hellauer and Horbach, 2024). In the EGU journals, all peer reviewer comments are immediately published, and they are subsequently archived as part of the interactive discussion, in which the referees, editors, authors and the scientific

community can participate. Therefore, we refer to it as 'public peer review' in the context of the EGU journals, which is arguably one of the longest practiced, best established and most successful forms of open peer review.





It seems that the importance of proper archiving and citability for manuscripts and reviewer comments has been overlooked in the open peer review experiments of various publishers and societies. For example, the American Geophysical Union (AGU) started experimenting with public peer review in 2008, building on the success of the EGU interactive OA journals, which were at that time already very well established in the global geoscience community. Unlike EGU, however, AGU did not offer permanent archiving, but deleted the discussion papers and interactive comments after public review and final acceptance or rejection of a manuscript (Albarède, 2009). This line was also followed by the Journal of Advances in Modeling Earth Systems (JAMES), which had initially adopted the interactive OA publishing concept of ACP, but did not maintain the archiving of comments in their discussion forum (JAMES-D), was taken over by AGU, and eventually abandoned open peer review (Pöschl, 2012). The 2006 'peer-review trial and debate' in Nature (Nature Editorial, 2006) was also not successful, because neither authors nor their colleagues and readers had much of an incentive to participate in the public discussion (Pöschl, 2012). Articles were posted in a public discussion forum while they underwent closed peer review with non-public referee comments. A very high fraction (93%) of all papers were rejected, not because of a lack of scientific quality but because they were not deemed sufficiently exciting for the interdisciplinary audience of the magazine. For the rejected manuscripts, the previously published comments would become inaccessible. The incomplete documentation of the scientific discourse, which was an inherent result of deleting discussion papers and comments, undermined several key aspects of the public discussion and peer review, including the documentation of controversial scientific innovations or flaws, public recognition of commentators' contributions, and deterrence of careless submissions. Such differences may appear subtle at first sight, but they may explain why several other trials of open peer review were much less successful than the approach of ACP/EGU. After all, most scientists do care what happens to manuscripts and comments, in which they have invested substantial effort, i.e., if their results and opinions voiced in a public review and discussion process remain traceable or not.

Already in 1996, the *Journal of Interactive Media in Education (JIME)* was launched, facilitating in a first stage of the publication process a 'private open peer review' between authors and eponymous reviewers. In a second stage, the public could comment on the author and reviewer comments, after which the editor advised the authors on the revision of their manuscript (Buckingham Shum and Sumner, 2001). Finally, the editor decided which, optionally edited, parts of the interactive discussion were published alongside the final paper. This interactive review concept, however, does not seem to be applied in JIME any longer after its relaunch in 2011 (Weller, 2012). The journal *Electronic Transactions on Artificial Intelligence (ETAI)*, launched in 1997 and discontinued as of 2006, introduced a different form of an interactive two-stage publication process (Sandewall, 2012; Hachani, 2015): In a first stage, the scientific community could publicly discuss the article. In a second stage, designated peer reviewers provided reports that were available to the authors and editors only, but not to the public. In both *JIME* and *ETAI*, the discussions were restricted to interactions between specific groups, i.e., only within the scientific community or between referees and authors, respectively, limiting the overall transparency of the review process and the traceability of the evolution of the scientific manuscript.

A variant of interactive and public peer review has been explored by *eLife*, an OA journal in the biomedical and life sciences, launched jointly in 2011 by the Howard Hughes Medical Institute, the Max Planck Society and the Wellcome Trust (Schekman et al., 2012). In the initial years of *eLife*, there was a non-public interactive discussion between reviewers and authors. As of 2021, this discussion is open to the public with full record of all reviewer comments and author responses (Eisen et al., 2020), similar to the concept as applied in the EGU journals (Figure 3). The review process in *eLife* was concluded with a public editor decision to accept or reject the paper. This latter step has been abandoned as of 2022; instead, the review process concludes with an editorial 'eLife assessment', implying that the public communication among authors, reviewers and editors is sufficient for readers to judge the research quality (Eisen et al., 2022). However, this concept has led to major controversy among the *eLife* editors (Else, 2022; Abbott, 2023) and to discussions about the interconnections between peer review, editorial policies and indexing by the Web of Science (Brainard, 2024a, b; eLife, 2024; Stern, 2024; Barbour et al., 2025; eLife, 2025).

As a compromise between traditional closed peer review and public interactive peer review, some publishers and journals, such as the medical and biological journals by *BioMed Central*, as of 2001, provide a record of the pre-publication history of the scientific exchange between the reviewers, editors and authors after the publication of a final peer-reviewed paper (Cosgrove and Flintoft, 2017). As of 2020, more than 500 journals publish peer reviewer reports alongside the published paper, either immediately or as post-publication review history; their publication may be mandatory or upon approval by the authors and/or reviewers (Wolfram et al., 2020).



Table 2 lists several of such journals, illustrating the pioneering role of EGU in public peer review. In later years, some publishers offered to publish reviewer reports for journal families or series, e.g. PLOS, Madison (2019); Elsevier (Bravo et al., 2019; Justman, 2019), IOP publishing (Banks, 2019), Wiley (Moylan et al., 2020) and Sage Publications (Sage, 2021).

**Table 2.** Selection of journals disclosing peer review reports (optional/mandatory, after/during review) sorted by the year, in which this feature was introduced.

| Journal                           | Publisher                   | Since      | Reference                              |
|-----------------------------------|-----------------------------|------------|----------------------------------------|
| ЛМЕ                               | The Open University         | 1996       | Buckingham Shum and Sumner (2001);     |
|                                   |                             |            | Pöschl (2012)                          |
| ETAI                              | Linköping University Elec-  | $1997^{1}$ | Pöschl (2012); Sandewall (2012);       |
|                                   | tronic Press                |            | Hachani (2015)                         |
| Atmospheric Chemistry and Physics | EGU/Copernicus              | 2001       | Pöschl (2004, 2012); Pöschl and Koop   |
|                                   |                             |            | (2008)                                 |
| BMC Public Health                 | BioMed Central <sup>2</sup> | 2001       | ReimagineReview, Moylan et al. (2014)  |
| Biogeosciences                    | EGU/Copernicus              | 2004       | Pöschl (2012); Dingwell et al. (2011)  |
| Climate of the Past               | EGU/Copernicus              | 2005       | Pöschl (2012); Dingwell et al. (2011); |
|                                   |                             |            | Wolff et al. (2011)                    |
| Ocean Science                     | EGU/Copernicus              | 2005       | Pöschl (2012); Dingwell et al. (2011)  |
| Biology Direct                    | BioMed Central              | 2006       | Koonin et al. (2013)                   |

| Nature                               | Springer                         | 2006 | Nature Editorial (2006), trial period of < |
|--------------------------------------|----------------------------------|------|--------------------------------------------|
|                                      |                                  |      | 6 months                                   |
| The Cryosphere                       | EGU/Copernicus                   | 2007 | Pöschl (2012); Dingwell et al. (2011)      |
| Economics e-Journal                  | Kiel Institute, ZBW <sup>3</sup> | 2007 | IFW Kiel                                   |
| Atmospheric Measurement Techniques   | EGU/Copernicus                   | 2008 | Pöschl (2012); Dingwell et al. (2011)      |
| Geoscientific Model Development      | EGU/Copernicus <sup>4</sup>      | 2008 | Pöschl (2012); Dingwell et al. (2011);     |
|                                      |                                  |      | GMD Executive Editors (2019)               |
| EMBO journal                         | EMBO Press                       | 2008 | Pulverer (2010)                            |
| JAMES                                | AGU                              | 2008 | Pöschl (2012)                              |
| Geochemistry, Geophysics, Geosys-    | AGU                              | 2009 | Albarède (2009)                            |
| tems; Global Biogeochemical Cycles;  |                                  |      |                                            |
| JGR—Earth Surface; JGR—Planets;      |                                  |      |                                            |
| Radio Science                        |                                  |      |                                            |
| Semantic Web Journal                 | IOS Press                        | 2010 | Janowicz and Hitzler (2012)                |
| PeerJ journals                       | PeerJ Publishing                 | 2013 | Wang et al. (2016)                         |
| Life                                 | MDPI                             | 2014 | Rampelotto (2014)                          |
| Royal Society Open Science           | The Royal Society Publishing     | 2014 | Royal Society Publishing                   |
| Journal of Negative Results in       | BioMed Central/Springer          | 2014 | Shanahan and Olsen (2014)                  |
| BioMedicine                          |                                  |      |                                            |
| Nature Communications                | Springer Nature                  | 2016 | Nature Editorial (2015, 2016, 2022)        |
| Educational Philosophy and Theory    | Taylor & Francis                 | 2016 | Peters et al. (2023)                       |
| European Journal of Neuroscience     | Wiley                            | 2016 | Bolam and Foxe (2017)                      |
| SciPost Physics                      | SciPost/arXiv                    | 2017 | Caux (2017)                                |
| Genome Biology                       | Springer Nature                  | 2017 | Cosgrove and Flintoft (2017)               |
| The Plant Cell                       | Oxford Academic                  | 2017 | Merchant and Eckardt (2016)                |
| Sci                                  | MDPI                             | 2018 | Abdin et al. (2021)                        |
| AGU Advances                         | AGU/Wiley                        | 2019 | Trumbore et al. (2020)                     |
| RSC Chemical Biology                 | Royal Society of Chemistry       | 2020 | RSC News (2020)                            |
| eLife                                | eLife Sciences Publications      | 2021 | Eisen et al. (2020)                        |
| ACS Central Science; The Journal of  | American Chemical Society        | 2021 | Garakyaraghi et al. (2021)                 |
| Physical Chemistry Letters           | (ACS)                            |      |                                            |
| The Journal of Neuroscience          | Society for Neuroscience         | 2023 | Kastner (2023)                             |
| European Journal of Higher Education | Taylor & Francis                 | 2023 | Seeber et al. (2023)                       |
| Development                          | The Company of Biologists        | 2024 | Briscoe and Brown (2024)                   |
| Molecular Human Reproduction         | Oxford Academic                  | 2024 | Boiani and Duncan (2024)                   |
|                                      |                                  |      |                                            |





While the post-publication of the peer review history does not allow for participation of other members of the scientific community in an interactive discussion with authors and referees, it provides at least evidence of the existence and the rigor of the peer review process. Such disclosure of reviewer comments upon publication of final papers should be regarded as a minimum standard for OA publishing, in order to counteract low scientific standards of (semi-)predatory and fraudulent journals that are solely motivated by the publishers' financial interests. The mere post-publication of reviewer reports, however, inherently leads to a kind of bias and loss of information because only the reports of finally accepted papers are shown, whereas the reviews for rejected manuscripts are lost. In addition, deleting the discourse on papers that do not ultimately result in journal publication diminishes the educational value of open peer review, as these examples also provide important learning opportunities and orientation for all involved parties.

Full transparency during the review process may be only achieved by not only publishing reviewer reports but also reviewer identities. Some studies suggest that this could result in fewer reviewers being willing to take on the task (van Rooyen et al., 1999; Fox, 2021) and lead to less critical reviewer reports, in particular from early career researchers who may feel intimidated about publicly criticizing more experienced colleagues (Rodríguez-Bravo et al., 2017). However, other studies did not find any significant evidence that open identities limit criticism (van Rooyen et al., 2010; Ross-Hellauer and Horbach, 2024). In the EGU journals, a significant number of referees voluntarily reveal their names (on average 19% (10 - 63%), Section 3.3, Table S2). This number is higher than in journals with closed peer review (~6%, Fox (2021)), demonstrating self-regulation of EGU's public peer review, in which reviewers take pride in and appreciate the acknowledgment they receive inherently for their publicly available, citable reports. In addition, referee reports can be entered to platforms like ORCID or the Web of Science Reviewer Recognition tool (formerly 'Publons', an independent platform (2012 - 2017, acquired by Clarivate in 2017), providing additional recognition for these scientific contributions.

Nature reported on the EGU approach (Gura, 2002), performed an open peer review trial (Nature Editorial, 2006), and recently announced that it will publish all reviewer reports for new papers, making mandatory what had been optional since 2020 (Nature Editorial, 2025):

"Since 2020, Nature has offered authors the opportunity to have their peer-review file published alongside their paper. Our colleagues at Nature Communications have been doing so since 2016. Until now, Nature authors could opt in to this process of transparent peer review. From 16 June, however, new submissions of manuscripts that are published as research articles in Nature will automatically include a link to the reviewers' reports and author responses. It means that, over time, more Nature papers will include a peer-review file. The identity of the reviewers will remain anonymous, unless they choose otherwise—as happens now. But the exchanges between the referees and the authors will be accessible to all. Our aim in doing so is to

<sup>&</sup>lt;sup>1</sup> Journal discontinued in 2002; <sup>2</sup> BioMed Central was acquired by Springer Science+Business Media in 2008 (Cockerill, 2008);

<sup>&</sup>lt;sup>3</sup> The journal is owned by the publisher De Gruyter since 2020.

<sup>&</sup>lt;sup>4</sup> See Table A1 for a list of all EGU/Copernicus journals that were launched later.

open up what many see as the 'black box' of science, shedding light on how a research paper is made. This serves to increase transparency and (we hope) to build trust in the scientific process.

As we have written previously, a published research paper is the result of an extensive conversation between authors and reviewers, guided by editors. These discussions, which can last for months, aim to improve a study's clarity and the robustness of its conclusions. It is a hugely important process that should receive increased recognition, including acknowledgement of the reviewers involved, if they choose to be named. For early-career researchers, there is great value in seeing inside a process that is key to their career development. Making peer-reviewer reports public also enriches science communication: it's a chance to add to the 'story' of how a result is arrived at, or a conclusion supported, even if it includes only the perspectives of authors and reviewers. The full story of a paper is, of course, more complex, involving many other contributors.

Many people think of science as something fixed and unchanging. But scientific knowledge evolves as new or more-nuanced evidence comes to light. Scientists constantly discuss their results, yet these debates are not contained in research papers and often remain unreported in wider science-communication efforts. The COVID-19 pandemic provided a brief interlude during which much of the world got to see how research works, almost in real time. It's easy to forget that, right from the start, we were continuously learning something new about the nature and behaviour of the SARS-CoV-2 virus. On television screens, in newspapers and on social media worldwide, scientists were discussing among themselves and with public audiences the nature of the virus, how it infects people and how it spreads. They were debating treatments and prevention methods, constantly adjusting everyone's knowledge as fresh evidence came to light. And then, it went mostly back to business as usual. We hope that publishing the peer-reviewer reports of all newly submitted Nature papers shows, in a small way, that this doesn't need to remain the case. Nature started mandating peer review for all published research articles only in 1973 (M. Baldwin Notes Rec. 69, 337–352; 2015). But the convention in most fields is still to keep the content of these peer-review exchanges confidential. That has meant that the wider research community, and the world, has had few opportunities to learn what is discussed. Peer review improves papers. The exchanges between authors and referees should be seen as a crucial part of the scientific record, just as they are a key part of doing and disseminating research."

This move is a major improvement for which the EGU interactive OA publishing approach and related initiatives aimed at opening up the review process and disclosing reviewer reports (Table 2) have paved the way during the past decades.

# 2.3 Publishing formats and platforms

# 2.3.1 Preprints


The idea of sharing non-peer reviewed manuscripts, nowadays called 'preprints', within scientific communities reaches back several decades: In the 1960s, the National Institute of Health circulated manuscripts by regular mail within 'Information Exchange Groups' (Green, 1964; Cobb, 2017). At the same time, researchers in the Soviet Union were encouraged to deposit their papers on *VINITI* for efficient dissemination (Hammarfelt and Dahlin, 2024). Similarly, in the 1970s, Ginsparg (1994)

distributed manuscripts on physics- and mathematics-related topics prior to publication, which eventually led them to launch the first preprint server *arXiv* in 1991. By now the same concept is applied to numerous other discipline-specific servers<sup>2</sup>.








In parallel to arXiv, publishers launched more-interdisciplinary preprint servers, e.g., SSRN (1994) (acquired by Elsevier in 2016), *Nature Precedings* (2007 - 2012) by the Nature Publishing Group, *Preprints.org* (2020) by MDPI, *ESSOAr* (2022) by Wiley/AGU, *Research Square* (2023) (acquired by Springer Nature in 2022) or by non-profit organizations such as *WikiJournal Preprints* (2023). To facilitate access and visibility of scientific publications in developing countries, regional initiatives started even before the official open access initiatives (Basilio, 2023). They include the Scientific Electronic Library Online (SciELO, 1997) of OA journals in Latin America and South Africa that was later complemented by SciELO preprints (2020). To date, the adoption and utilization of preprints greatly varies across regions and scientific disciplines (Rzayeva et al., 2025). Other research communities started promoting the advantages and benefits of preprints such as 'Accelerating Science and Publication in Biology' (ASAPbio, 2017). In 2001, EGU/Copernicus started the interactive journal discussion forum for 'Atmospheric Chemistry and Physics' to share papers in an interactive discussion and peer review; this concept is by now adapted in the 18 newer EGU journals (Table A1) and complemented by the interdisciplinary preprint repository *EGUsphere* (2022).

Generally, preprints are indexed in common databases (GoogleScholar, Scopus, since 2017) and, thus, are citable as non-peer-reviewed publications, sometimes also referred to as gray literature. Both benefits and shortcomings of preprints became particularly obvious during the COVID-19 pandemic: On the one hand, the quick dissemination of novel scientific evidence was crucial to collaboratively develop solutions to the global crisis. On the other hand, the public and media are often not fully educated about the non-peer-reviewed, sometimes preliminary status of the results presented in preprints and how to interpret the published information, potentially leading to premature assessments or conclusions (Fraser et al., 2021; Drury, 2022; Schultz, 2023; Fleerackers et al., 2024; Brainard, 2025a).

To overcome the lack of quality assurance for preprints, Boldt (2011) proposed to extend *arXiv* by a peer review model, similar to journals; however, this idea did not succeed in the suggested form. Later, similar concepts were termed 'publish-then-review model' or peer review of preprints in 'overlay journals', in which manuscripts on preprint servers undergo peer review (Tennant et al., 2017; Rousi and Laakso, 2022) (Section 2.2). An example for successful overlay journals combining *arXiv* preprints with an interactive OA publishing concept are the *SciPost Journals* published since 2016 in physics and other fields, with a structure similar to that in the discussion forums of the EGU journals and *EGUsphere* (Sections 3 and 4.2).

When ACP was launched in 2001 as EGU's first interactive scientific OA journal, preprints posted for public review and discussion were labeled as 'discussion papers'. This labeling indicated that these papers already passed some form of basic scientific access review by an editor, optionally supported by referees, before they were accepted for public review and discussion in 'Atmospheric Chemistry and Physics Discussions (ACPD)', the discussion forum of ACP. It was clearly indicated on all relevant web pages and through watermarks in the papers' PDF files that discussion papers are not fully peer reviewed scientific articles as opposed to final journal papers. For clarification across and beyond the field of geosciences, this was also expressed in an official 'EGU Position Statement on the Status of Discussion Papers Published in EGU Interactive Open

<sup>&</sup>lt;sup>2</sup>e.g., biorxiv.org (2013), socopen.org (2016), psyarxiv (2016), chemrxiv.org (2017), paleorxiv.org (2017), LawArXiv (2017-2021), eartharxiv.org (2017), metaArxiv (2017), medrxiv (2019), techrxiv (2020); all links last accessed 24 Jan 2025.

Access Journals' (EGU News, 2010). Since ACP's launch, the community has readily embraced and accepted this distinction between discussion papers and final journal papers. In fact, ACP and EGU's interactive OA publishing initiative might not have succeeded without this clear labeling - and at that time even quite distinct typesetting - of discussion papers/preprints prior to being publicly exposed. These measures and the introduction of digital object identifiers (DOIs) and related labels effectively dispelled widespread initial concerns and misperceptions about plagiarism of preprints and introduced the novel concept of preprints with public peer review, many years prior to the launch of traditional preprint servers in the Earth sciences (Pourret et al., 2021). By now, most publishers accept submissions of previously preprinted manuscripts for peer review and possible subsequent publication in a peer-reviewed journal.

To date, the more general terms 'preprint' and 'preprint repository' for the different types of preprints (with and without peer review) on *EGUsphere* (Section 4.2) largely replace the terms 'discussion paper' and 'discussion forum' across EGU and in the wider (geo)scientific community. This adaption simplifies the terminology in the (geo)scientific publishing landscape and on the web page structures of EGU and its publisher Copernicus. We propose, however, that the term discussion paper should not be fully abandoned as a useful label for preprints that underwent a scientific preselection process by an access review by a journal editor (Section 3.2) and are undergoing full, public peer review (Section 3.3), as opposed to other traditional, standalone preprints that have not undergone and may never undergo any substantial scientific quality assurance. Such distinction is valuable for our understanding of an epistemic web as a dynamic and interconnected space to trace the creation, sharing and construction of knowledge.

# 2.3.2 Open-access publication platforms with transparent, public peer review





During the past century, different publishing formats have emerged that allow for distributing work prior to publication in scientific journals and to receive feedback by peers. An early example of discussions of unpublished work are the *Faraday Discussions*, a journal launched in 1947 by the Royal Society of Chemistry. It publishes research papers presented at 'Faraday Discussions Meetings', together with a record of the questions, discussion and debates that had occurred during the meeting. However, the discussion is limited to the meeting participants only and thus greatly differs from today's public, interactive discussions on online platforms.

The potential of the internet for interactive discussions among much wider communities was recognized by Harnad (1992) who implemented the concept of 'scholarly skywriting' into the OA journal *Psycoloquy* (discontinued as of 2002), which allowed authors to solicit feedback by peers from all over the world on their new ideas and findings. In 2002, Berkeley Electronic Press (Bepress), in collaboration with the California Digital Library, launched the eScholarship Repository to share 'working papers' in the humanities and social sciences to allow for soliciting feedback before formal peer-reviewed publication<sup>3</sup>. Within the Economics community, several platforms and outlets exist(ed) for the early sharing of papers. The meta-data and abstracts of non-peer-reviewed working papers, published by individual research institutions, were compiled within the journal 'Abstracts of Working Papers in Economics' (*AWPE*, Cambridge University Press; discontinued in 2004) that was launched in 1986 and enabled researchers to discover new work from over 70 research centers. In addition, RePEc (*Research Papers in* 

<sup>&</sup>lt;sup>3</sup>Bepress was acquired by Elsevier in 2017 (MacKenzie, 2017)

*Economics*) was launched in 1997 as an OA platform where researchers and institutions share working papers, articles and related outputs.

This early concept of an online interactive discussion in a scientific journal has further developed since then and is practiced nowadays on numerous scientific publishing platforms, as outlined below and in scientific journals, including the EGU journals. In 2012, the OA publisher F1000 launched the open research publishing platform *F1000 Research* for the peer review of preprints. Authors submit their manuscript for immediate posting and suggest potential reviewers. Upon a sufficient number of favorable reviewer recommendations, the paper status is considered final in order to be indexed in bibliographic databases (Scopus etc). Authors can upload updated versions of their manuscript at any time, even after indexing. In the same year, *PeerJ* was launched applying the same sequence of manuscript posting and peer review (PeerJ, 2012; Binfield, 2014). *F1000* and *PeerJ* initially differed in their business model; the former was fully financed through article processing charges (Lawrence, 2012), and the latter applied a membership-based model for authors which was extended in 2016 to allow for payments of individual articles. Both *F1000* and *PeerJ* were acquired by the commercial publisher Taylor&Francis in 2020 and 2024, respectively (Taylor & Francis News, 2020; PeerJ Blog, 2024).

Since 2012, several other *F1000*-managed platforms were launched, e.g., *Wellcome Open Research (2016)*, *Gates Open Research (2017)* and *Open Research Europe (2017)*, the latter of which is open to all scientists funded by the European Horizon2020 program. As of 2025, the Bill & Melinda Gates Foundation makes the publication of preprints mandatory when they report on studies that were funded by the foundation, followed by optional peer review on their preprint platform *VeriXiv (2024)*. The popularity of these platforms apparently stems from their OA and publish-then-review concept without charges, with peer review rigor being a secondary criterion (Whitfield, 2012; Kirkham and Moher, 2018). These priorities frequently lead to concerns about the quality assurance on such OA publishing platforms despite the fully transparent and public peer review. In addition, Ross-Hellauer et al. (2018) raised ethical concerns with regards to funder-supported publishing platforms due to potential biases in the selection of published research results. Moreover, they warned that the quality and reputation of such platforms may decrease, if authors considered such platforms as an inferior choice and rather submit their best papers to highly prestigious journals.

Non-profit initiatives triggered the creation of funder-independent platforms. For example, the French-led *Peer Community in* (PCI, 2016) facilitates open peer review of preprints deposited on the *Episciences* platform (CCSD, 2017), in the French *'Hyper Article en Ligne'* repository (HAL, 2001) or on several other preprint servers (OSF preprints, *PaleorXiv*, *EcoEvorxiv*, *AfriArxiv*, *SocArXiv*, and *bioRxiv*). Preprints posted on these servers can then be linked to one of 19 thematic PCIs for open peer review. Upon acceptance of the paper by an editor, it can be either published at no-cost in the *Peer Community Journal* (PCI, 2016), or transferred to a 'PCI friendly journal' for potential publication, possibly without further peer review. In 2017, about 50 preprints were submitted and also recommended; these numbers increased to 518 and 240, respectively, in 2024, with each preprint receiving 2 - 3 reviews on average. In total, about 1800 preprints were linked to the PCI platform, 830 papers were recommended for publication in either the Peer Community Journal or in an PCI friendly journal (about 50% each) (PCI Facts & Figures, 2024).

The platform *Review Commons (2019)* was created as a joint initiative by the European Molecular Biology Organization (EMBO) and the non-profit initiative ASAPbio and allows for open discussion and peer review of external preprints posted on *bioRxiv, medRxiv* and, since 2022, *SciELO preprints* (Lemberger and Pulverer, 2019). The authors may either transfer their peer-reviewed preprints to a *Review Commons* partner journal<sup>4</sup>, or just leave their archived preprints on the preprint server accompanied by the peer review reports. The same concept of preprint peer review is applied to submissions to the *JMIRx* journals that require posting a preprint as JMIR preprint or on *bioRxiv* or *medRxiv* which then undergoes discussion in a *PREreview* journal club or peer review by a *Plan-P* accredited service (JMIR, 2022). The *JMIRx* platform acts as an overlay journal and authors can select the journal, in which their preprint should be peer reviewed. On the same platform, editors can select preprints for potential peer review in their journals (Eysenbach, 2019). In 2022, the journal *Society (2022)* by the Microbiology Society was transitioned into an open publishing platform where all versions of an article are posted as preprints together with peer reviewer reports. The review process concludes with an editor decision to accept a final version of the paper that is indexed in common data bases. It recently joined *Sciety*, a platform created by eLife that provides a compilation of preprints that were peer reviewed on different platforms, including Biophysics coLab, eLife, preLights, Review Commons, ASAPbio and PeerJ.

Many publishing platforms have in common that authors suggest their peer reviewers to be nominated without further editorial selection. Such 'review by endorsement' was suggested to potentially make peer review more efficient (Velterop, 2015). In 2024, F1000 introduced an editorial-led-peer-reviewer-selection (F1000, 2024), mainly to speed up the review process that was often delayed by having to verify author-suggested reviewers. The platform QEIOS (2019) relies entirely on reviewers selected by an artificial intelligence (AI)-based tool to identify preprint commentators. Similar as on other (e.g., F1000-managed) platforms, the review process concludes with recommendations by the reviewers only, without any final editor decision. Although QEIOS has been referred to as an innovative new journal (Columbia University, 2022), its lack of a final editor decision does not adhere to the selection criteria for scientific journals as defined by the Web of Science (Clarivate, 2024). The examples above represent (more or less) successful platforms for peer review of preprints for potential journal publication. Several of these platforms include aspects of the concept as applied in the EGU journals and their related discussion forums and preprint repository EGUsphere (Section 4).

Moreover, publishing platforms and preprint servers offer great opportunities for various additional purposes, including the educational value for younger scientists to discuss, evaluate and constructively criticize scientific publications as recognized by several communities. Examples include *PRElights* (2018), a community initiative supported by *The Company of Biologists*, where early career scientists organize the discussion and highlighting of external preprints. Similarly, the *PREreview* (2017) initiative organizes trainings for early career scientists, including feedback to preprint authors and collaboratively written reviewer reports on preprints in connected overlay journals. Richter et al. (2023) proposed that such journal clubs, which carry out community-based peer review might possibly help to alleviate the burden on traditional journal-based peer review by expanding the pool of reviewers, providing timely feedback and fostering a collaborative review process.

<sup>&</sup>lt;sup>4</sup>Including journals published by EMBO Press, eLife, The American Society for Cell Biology (ASCB), The Company of Biologists, Rockefeller University Press, Public Library of Science (PLOS); all links last accessed on 24 Jan 2025.

The *PubPeer* platform, launched in 2012, differs from the concepts of the various publishing platforms since it only allows for discussion and review of peer reviewed journal articles published elsewhere. It is primarily used to point out flaws or scientific fraud. Readers of the original article, however, may not be aware of these critiques because the journal websites usually do not provide a link to the discussion on *PubPeer*. To address this gap, a notification tool has recently been developed to establish links between the platform and the original publication (Singh Chawla, 2024).

The number of different publishing platforms (with varying rigor and procedures) and, hence, also the number of peer-reviewed preprints has sharply increased over the last few years (Brainard, 2022; Avissar-Whiting et al., 2024). Sondervan et al. (2022) and Lutz et al. (2023) provide overviews of the broad range of features, including different forms of public peer review that are offered by a few of them. They also differ in the workflows how preprint metadata are stored, transferred and eventually linked to journal articles (Alves et al., 2024). Their main common feature is the fast publication and transparency in peer review, also termed 'publish, review, curate' that has recently been proposed by the OA initiative *Plan S* as the essential standard for all OA scientific publications (Liverpool, 2023). This concept has been applied in *Atmospheric Chemistry and Physics* since 2001 and in all other EGU journals that were launched subsequently.

# 475 2.4 Bibliometric indicators of visibility and quality: Traditional measures and new opportunities





The quality and importance of scientific journals is often judged by means of the journal impact factor (JIF), which denotes the ratio of citations in a given year to citable articles published in a particular journal during the preceding two years. The JIF was initially developed as a guideline for librarians to select the most popular journals within a discipline (Garfield and Sher, 1963). Using the JIF as an indicator for the quality or impact of individual papers in a given journal is, thus, highly questionable and potentially even misleading because the JIF

- does not give any direct information on the quality or impact of an individual article (Seglen, 1997; Simons, 2008; Nature Editorial, 2013; Casadevall and Fang, 2014).
- was shown to be determined predominantly by only a small number of papers (e.g., highly cited review articles), whereas
  most articles belong to the 'long tail' that have citation counts much lower than the JIF suggests (Triggle and Triggle,
  2017; Antonoyiannakis, 2020).
- is prone to be enhanced by way of coercive journal citation malpractices, such as excessive self-citations (citation stacking) (Kulczycki et al., 2021; Oviedo-García, 2021; Siler and Larivière, 2022) or citation cartels (Kojaku et al., 2021).
- is particularly sensitive to the number of *citable* items in a journal. The denominator in the JIF calculation, normally includes only papers that are classified as 'articles' but excludes editorials, news items, commentaries, letters to the editor etc. which are, however, counted in the numerator (Hernán, 2009; McVeigh and Mann, 2009; Manley, 2022).
- may be artificially inflated by commentaries that lack genuine scientific findings and conclusions, and can be AI-generated. These commentaries, often classified as opinion pieces, receive disproportionate weighting in the JIF calculation and are frequently encouraged by journals to include citations to their own articles (Joelving, 2024).

Given these shortcomings, the OA publisher PLOS refrains from reporting JIF and related journal indicators and limits their reporting to article-based measures (Public Library of Science (PLOS)). Similarly, EGU journals state on their start pages that the journals are indexed in the Web of Science, Scopus, Google Scholar, etc. However, the journal metrics are not prominently advertised because they do not describe importance, impact, or quality of a journal if used in isolation. Therefore, it is explicitly stated on the journal pages that the use of journal metrics is discouraged due to widely recognized limitations.

Several bibliometric measures alternative to the JIF have been suggested to account for discipline- or topic-specific citation statistics (Bornmann and Haunschild, 2016; Bornmann and Marx, 2016). The limitations of such bibliometric measures for evaluating the work of individual scientists have been acknowledged over the past decade by research funders, institutions and other entities. This recognition has prompted proposals for more equitable and comprehensive assessments of scientific influence and societal impact, accounting for the entire range of research outputs, practices and scholarly activities (Pourret et al., 2022; Triggle et al., 2022; Trueblood et al., 2025). This notion is expressed in several international declarations, including the *Declaration of Research Assessment* (DORA, 2012), the *Leiden Manifesto* (2015) and the *Coalition for Advancing Research Assessment* (CoARA, 2022)<sup>5</sup>.

OA publishing with interactive public discussion provides further opportunities to make scientific impact beyond traditional article-level metrics. In addition to citations, *Altmetric* details are commonly reported as a measure of public engagement and feedback on scientific publications (Priem et al., 2010; Shuai et al., 2012; Taylor, 2023) as they account for data from social and traditional media, blogs and online reference managers (Altmetric, 2023).

The 'open access advantage' that was initially only shown in terms of higher citation counts of OA articles as compared to those in pay-walled publications (Eysenbach, 2006; Xie et al., 2021), can be extended to differences in *Altmetrics* (Fu and Hughey, 2019; Clayson et al., 2021; Vadhera et al., 2022; Chen et al., 2024; Cheng et al., 2024). OA articles are not only more frequently cited in the Wikipedia encyclopedia (Teplitskiy et al., 2017) and more visible to the public via social media (Schultz, 2021), but they are also more readily accessible to policy makers (Xu and Zong, 2024). The aforementioned indicators focus on the impact of scientific papers and, thus, give recognition to their authors. However, also the participation in scientific discussions can be considered an intellectual contribution to the advancement of science. Therefore, citable peer reviewer and community comments on preprint servers and publishing platforms, as provided in the EGU journal discussion forums and on *EGUsphere*, should be considered valuable as they add to scientific discourse and debates and therefore are essential elements of the epistemic web of knowledge (Figure 1).







<sup>&</sup>lt;sup>5</sup>EGU signed both DORA and CoARA in October 2024 (EGU News, 2024b).

#### 3 Multi-stage open peer review with public discussion

For more than two decades, the traits of publishing platforms as described in Sections 2.3.1 and 2.3.2 have been successfully applied and combined in the EGU's multi-stage interactive publication model. In the following, we outline the steps from manuscript submission to potential final publication in the 19 EGU journals (Figure 3).

**Figure 3.** Schematics of the multi-stage, interactive peer review process as applied in EGU journals; solid arrows denote mandatory steps, dashed arrows are optional actions

# 525 3.1 Manuscript submission



During manuscript registration, authors provide a brief summary of their article along with a statement indicating its alignment with the journal scope. At this stage, authors select prescribed subject areas for the later editor calls. In addition, authors choose the manuscript type of their paper (Section 4.1.2) and optionally a special issue (Section 4.1.3). Once submitted, manuscripts undergo an initial basic technical check ('file validation') by the publisher's editorial support team, upon which editors of the matching subject area are informed of the submission by automatic emails and are invited on a first-come/first-served basis to handle the manuscript. If no editor agrees, editor calls are repeated several times, increasingly broadening the editorial subject areas, to finally all editorial board members. If these calls are unsuccessful, the executive editors decide on how to proceed: They either reject the paper or manually assign or nominate an editor with suitable expertise and/or with a low editorial workload. Manuscripts that do not find an editor during the automated calls are often found to be at the edge of the journal scope, for which no editor considers themselves an expert, or are weak papers not suitable for public discussion. In

EGU journals with relatively low numbers of submissions (ESurf, GC, GChron, NPG, SOIL, SE, see Table A1 for full journal names), every submission is assigned manually to an editor by an executive editor.

# 3.2 Access review





As part of their initial decision, the handling editor evaluates whether the paper fulfills the main review criteria, such as the *ACP review criteria*:

- Scientific significance: Does the manuscript represent a substantial contribution to scientific progress within the scope of the journal (substantial new concepts, ideas, methods, or data)?
- Scientific quality: Are the scientific approach and applied methods valid? Are the results discussed in an appropriate and balanced way (consideration of related work, including appropriate references)?
- Presentation quality: Are the scientific results and conclusions presented in a clear, concise and well-structured way (number and quality of figures/tables, appropriate use of English language)?

Editors are expected to make the initial decision by themselves, in particular if the paper falls into their core expertise. Editors of some journals (ACP, AMT, BG, WCD) can ask referees for a 'quick report' to aid their decision. The editor and referees may provide suggestions for minor/technical corrections (e.g., typos and clarifications); any revisions beyond those are not foreseen at the access stage and lead to rejection, possibly with the option of a resubmission of a suitably revised manuscript. In the case of resubmission, the authors are asked to explain how previous criticisms were addressed. As part of their decision, the editor confirms or adjusts the manuscript category; an adjustment does not imply a rejection but may require a change of the title to include the manuscript type (e.g., technical note, measurement report or opinion in ACP, Section 4.1.2). Reasons for rejection during the quick access stage may include the recommendation to transfer the paper to another Copernicus journal, which better fits the topic of the submitted manuscript. The rejection rate in EGU journals at the access stage is ~16% (in 2024) on average with some differences between journals (Figure A1), which is higher as compared to 10% in 2009. However, a comparison of rejection rates in journals of related disciplines in 2010 showed that EGU journals had significantly lower rejection rates at that time (Schultz, 2010). The relatively low rejection rates in EGU journals suggest a form of self-regulation by the public peer review process, which encourages authors to submit high-quality manuscripts for public discussion and to refrain from submitting poor manuscripts on a trial-and-error basis, possibly because authors may want to avoid receiving bad reviews for their papers in public.

# 3.3 Public discussion and peer review

After the quick access review and acceptance for public discussion, manuscripts are posted in the discussion forum of the particular journal and on *EGUsphere* (Section 4.2). A permanent DOI is assigned to the preprint and its status is indicated by the addition of [preprint] in the citation and by the DOI 'egusphere-year-number' (or previously '[journal]-year-number', discontinued as of the beginning of 2025). The duration of the public discussion phase depends on the journal and on the manuscript type and varies from 5 to 8 weeks for regular articles and 4 weeks for letter-style articles (Section 4.3.3). The

scientific community is informed about new preprints/discussion papers and invited to contribute to their public discussion via the respective journal website and the *EGUsphere* website, automated alerts (upon previous sign-up) and social media. At the beginning of the discussion phase, the editor nominates referees until at least two of them agree to provide a report. For the nomination, the editor can take advantage of various search tools provided by the publisher (Section 4.1.5); referees who provided input at the access stage are automatically re-nominated. The editor can select numerous referees initially to be successively called in the order of their choice, if one or several of them decline. To minimize the generation of unnecessary referee reports and conserve the valuable resource of referee time, the editor is notified once a sufficient number of referees has been secured. Nominated referees whose nomination deadline did not expire yet may still accept the nomination, unless it is manually terminated by the editor. If the quorum of 2 referee reports is not fulfilled during the primal discussion period, the discussion is automatically extended, and the editor is requested to (re)nominate additional referees. On average, each preprint receives 2 - 3 peer reviewer reports which are published (Figure A2). In addition, referees can send confidential comments to editors that are not publicly shared, allowing them to raise sensitive concerns.

Referees can optionally keep their identity anonymous ('single-blind review'), allowing them to provide critical comments without worrying about negative personal consequences, if authors or other scientists are dissatisfied with their comments. Independent studies have shown that the option of anonymity leads to more thorough and potentially more constructive reviewer comments (Khan, 2010; Shoham and Pitman, 2021). Concerns that anonymity removes the accountability of referees and, therefore, leads to hostile comments or unsubstantiated criticisms are outweighed when all reviewer comments are publicly posted and can be checked for relevance and credibility. The option to remain anonymous is reserved for referees nominated by and known to the editors.

Any member of the scientific community may post eponymous comments during the discussion phase. Such community comments contribute about 5% to all comments (3 - 17%, depending on the journal) and are about a factor of 10 less frequent than regular referee comments (Figure 4). Frequently, the discussions contain 20 or more comments from all involved parties (Table S2). For example, the discussion of the article by Hansen et al. (2016) with a total of 110 comments evolved among the first author, 2 referees, the editor and 26 additional members of the scientific community (Interactive discussion: Hansen et al., 2016). Statistics of the most commented papers in EGU journals show that these papers are often not regular research articles but opinion articles, review articles or peer-reviewed comments that may be controversial and motivate the community to contribute to the public discussion (Table S3). In many cases, the interactive discussion led to significant improvements of the paper such that several highly commented papers were finally selected as highlight articles. In other cases, however, the discussion led to the identification of major flaws in the paper so that either no revised manuscript was submitted or the revised manuscript was not accepted for final publication.

This distribution of referee, author, editorial and community comments in the interactive discussion is broadly in agreement with that in other journals with interactive platforms such as PLOS (Wakeling et al., 2019). There, most comments are made by authors or editors; attributed comments are mostly related to the publication process (language, typesetting, referencing) or to scientific or technical soundness. To stimulate scientific exchange between all parties, the authors of EGU journals are encouraged to interact with the commentators during the discussion phase, rather than just posting an author response to all

comments after the end of the public discussion. Executive editors may alter or remove comments that are inappropriate, personally insulting or scientifically not relevant. However, this has been applied to a negligible number of all comments (

**Figure 4.** a) Total number of comments on discussion papers/preprints of EGU journals since 2001; b) Number of comments per year. The distribution of comments in the individual journals in Table S2 (Supplemental information).

# 3.4 Peer review completion




The concept of ACP, as initially developed and still applied, foresees that the editor does not interfere between the two stages of the peer review process (Figure 3). An editorial decision immediately after the interactive discussion may bias the peer review completion, for example, when the editor asks for specific changes before the authors respond to all referee and community comments and have a chance to upload a revised manuscript. Instead, ACP authors are always given the opportunity to revise their manuscript to demonstrate and enhance its quality before any editorial decision. After receiving critical feedback during the public discussion, scientists are expected to decide on their own on how to address comments and concerns. Only if necessary, e.g., if they are uncertain about whether revising their manuscript in a specific way would lead to a publishable paper, they may seek guidance from the editor. Generally, ACP authors appreciate this independence to improve their manuscript without further editorial interference. In fact, their revisions after public discussion frequently even exceed the requests and suggestions by the referees.

At present, 10 EGU journals (ANGEO, BG, CP, ESD, GC, GChron, HESS, NHESS, SOIL, TC) deviate from this original concept by imposing a mandatory editor decision after the authors have responded to the discussion comments, but prior to uploading any revised manuscript. This 'post-discussion editor decision' is often felt disrupting when authors prepare their response to the referees and the revised manuscript simultaneously. In journals without post-discussion editor decision (ACP, AMT, ESurf, GI, GMD, NPG, OS, SE), authors are always given the chance to revise and improve their manuscript in response to the discussion, while in the other journals, papers may be rejected at this stage, even though authors potentially may have

been able to revise their manuscript satisfactorily. Such rejections prior to manuscript revision may (in part) explain the higher average post-discussion rejection rates ( $\sim$ 10% vs  $\sim$ 4%, Figure A1f) and longer processing times (Section 4.1.6) in these journals as compared to those that forgo early editor interference.

#### 3.5 Optional re-review by referees

When the authors upload their documents for peer review completion, which includes a point-to-point response to all comments as well as the revised manuscript (including a version with changes tracked), the handling editor is automatically informed. The editor makes a decision with or without consulting with previous or new referees; this decision may include the request for further revisions. If required, the process of re-review and revision can be iterated multiple times. However, in the interest of processing times, the iterations should be limited and terminated, if it becomes clear that further revision will not result in a paper version that may eventually be acceptable for publication.

#### 3.6 Final editor decision





After the revisions by the authors, the editor makes their final decision on acceptance or rejection. In the case of acceptance, all editor and referee reports, manuscript versions and author responses that were prepared during the peer review completion stage are made public alongside the final journal paper. In case of rejection, the editor is expected to post a public editor comment, in which they explain the reason(s) for their decision and, thus, preempt appeals or requests for clarification by the authors. Such public editor comments should be posted also if an editor decision overrules important referee comments, or when referees had differing views. These comments help to explain the editor decision and give public acknowledgement to the contribution of the referees during the revision process. Thus, the active editor role in making decisions on the manuscript can be transparently tracked for all published papers. In other peer review models, in which such editor reports are not made publicly available, the extent to which editors may act only in a judicial role is intransparent (Tennant and Ross-Hellauer, 2020).

The low rejection rate of manuscripts after the discussion phase in EGU journals (Figure A1b, d, f) can be attributed to two main reasons: first, the number of initial submissions of deficient manuscripts is relatively low as authors hesitate to trigger negative reviews in the public review process; second, if major deficiencies are present, they are usually either identified during the access stage or sufficiently addressed by appropriate revisions. Rejected manuscripts and their preprint DOI are permanently archived and, thus, remain accessible. Final published journal papers receive a new DOI, reflecting the journal (in the format *journal-year-firstpage*). The preprint and the previous interactive discussions are linked to the final journal publication and are accessible both via the *EGUsphere* and journal websites.

As part of the final decision, the handling editor can select a paper as a highlight paper. Already at manuscript submission, authors may justify why they consider their paper to belong into the highlight category. However, any paper can become a highlight, even without the authors' proposition. In either case, the editor has to explain in a short statement how the paper fulfills the highlight criteria that include (i) important discoveries, or major advances in long-standing questions within the journal scope, or (ii) scientific advances of high interest that are accessible to the broad geoscience community or to the broader public and media. Building on the referee ratings and the justification by the handling editor, the journal executive editors make

the final decision whether a paper qualifies as a highlight article. A published highlight article is accompanied by an executive editor statement on the article website. This additional selection step by the executive editors was introduced in 2020 to achieve greater consistency in selection of highlights. In addition to the highlight selection as 'editor's choice' (Figure 5), articles of journal-specific manuscript types and letters qualify automatically as highlights (Sections 4.1.2 and 4.3.3).

**Figure 5.** The different routes of highlight article selection in EGU journals. All highlight articles are selected by executive editors (EE, or referred to as chief editors (CE) in some journals) and accompanied by an editorial statement alongside the final journal publication. Highlight articles by 'editor's choice' are selected after the full peer review. Some journal-specific manuscript (MS) types (e.g. ACP opinions) or letters Section 4.3.3, the MS category indicates already their highlight potential; however, they may be recategorized during the review process.

#### 4 EGU publishing platforms



All preprints and peer-reviewed journal articles allow immediate free and open access to full-text PDF, HTML and XML that are distributed with the *Creative Commons Attribution License CC BY 4.0*. This license enables the wide use and sharing of work while ensuring proper credit of the work to its authors who retain the copyright of their articles. This license should be preferred over those for papers in pay-walled journals where authors may self-archive their papers only in non-final forms (pre-final, final but not typeset or typeset) and after potential embargo times imposed by the publisher ('green OA', Section 2.1.2).

# 670 4.1 EGU journals






# 4.1.1 Statistics of submissions, preprints and published papers

An overview of all EGU journal titles and the year of their first publication applying the interactive publishing model is given in Table A1. Some journals were initially launched with closed access (ANGEO: 1983, NPG: 1994, NHESS: 2001) but were converted subsequently into OA journals in 2009, 2014 and 2004, and they introduced interactive, multi-stage public peer review in 2018, 2014 and 2013, respectively. From 2004 onward, all newly launched EGU journals have been full OA journals with multi-stage public peer review and interactive discussion from their start.

Figure 6 shows the number of submissions, preprints (discussion papers) and published papers in each journal (panels in the first and second rows) and the total of all EGU journals (third row), as well as the cumulative numbers since 2001 in the bottom row. In most journals, there is a continuous increase in the number of papers in each of the three categories. The downward trend in published papers for ANGEO (red lines in Figure 6f) is striking. Initially, the publication costs of ANGEO were largely covered by institutional subscriptions. In 2009, ANGEO was converted into an OA journal in 2009. As of then, authors had to pay individual APCs, which in many cases may not have been covered by institutional agreements (Section 4.1.7). This change in the financing model may have contributed to its decreasing submission rate.

The difference between number of submissions and the number of preprints in all EGU journals is on average 

**Figure 6.** Number of papers in EGU journals (2001 - 2024); first column (panels a, d, g, j): Submissions; second column (panels b, e, h, k): Preprints/discussion papers; third column (panels f, i, l): Journal publications. Top row: OA EGU journals that introduced the multistage peer review model prior to 2012 (Pöschl, 2012); second row: OA EGU journals that were launched later or introduced the multistage peer review model after 2012; third row: sum of papers for all 19 EGU journals. The dashed lines in panels c) and f) denote the period when journal were open access but did not yet apply the multistage peer review model (ANGEO: 2009 - 2018; HESS: 2004 - 2014; NHESS: 2004 - 2012); bottom row: cumulative numbers since 2001 for all EGU journals.

not fully established and accepted by other publishers. These retractions were initiated by authors after unfavorable reviews or rejections of their papers, so that they could be submitted to other journals; in later years, such papers are typically labeled as 'withdrawn', consistent with the terminology as used in other journals. The remaining 5 papers (0.03%) were retracted from ACP because of invalid results as identified after publication. These trends are in line with results from a comparison of different peer review models that revealed that during the public peer review, major flaws or fraud are usually caught and either are corrected by the authors during revision or result in a rejection (Horbach and Halffman, 2019). These retraction rates are lower by at least an order of magnitude as compared to those in journals that do not provide transparency of peer review (Fang et al., 2012; Van Noorden, 2023; Retraction Watch, 2025).





ACP was not only the first, but also is by far the largest EGU journal with  $\sim$ 800 published papers annually during the past decade (Figure 6a). While the number of submissions to ACP has somewhat increased since 2016, this increase is not fully reflected in the numbers of published preprints, because concomitantly the rejection rate has slightly increased from  $\sim$ 15% to 20% (Figure A1a). To illustrate the evolution of ACP in more detail, Figure 7 shows the trend in the distribution with regards to the *ACP subject areas*, up to two of which are selected by the authors during submission.

**Figure 7.** Percentage of manuscript submissions to *Atmospheric Chemistry and Physics* for each subject area (2009 - 2024). The subject areas isotopes, biosphere/atmosphere interactions and hydrosphere/atmosphere interactions are discontinued as of 2023, radiation as of 2025, whereas climate & Earth system was added in 2023.

The areas 'aerosols' and 'gases' clearly dominate, representing consistently > 70% of all submissions, whereas the areas 'clouds & precipitation', 'dynamics', 'radiation', 'isotopes', 'biosphere interactions' and 'hydrosphere interactions' received a minor share of submissions. Areas that describe interactions of the atmosphere with other Earth compartments (biosphere and hydrosphere) contributed 

**Figure 8.** Number of published ACP papers by manuscript category (2011 - 2024). a) Total number and the main category 'research article'; b) All other manuscript categories. Highlight articles are not a separate category but are shown for reference (see main text). ACP Letters and Opinions are highlight articles by default; articles in other manuscript categories may be highlighted as editor's choice (Section 3.6).

Also in 2020, ACP was the first EGU journal that introduced the manuscript category 'Letters' (Section 4.3.3). Manuscripts in this category and also articles in the category 'Opinions' in ACP, are expected to be ranked as highlight articles upon acceptance for final publication. Unlike highlights selected upon the editor's choice that can be papers of any manuscript category (Section 3.6), Letters and Opinions are designated by default to become highlight papers already at the submission stage. Therefore, the executive editors check the suitability for these two manuscript categories prior to the discussion stage ('Letter manuscript' and 'journal-specific MS type expected to become highlight' in Figure 5). The decrease in the number of highlight articles since 2020 (Figure 8b) is likely due to the introduction of an additional step in that year, in which the executive editors have to approve (or reject) the handling editor's highlight recommendation (Section 3.6). The proportion of highlight papers in the EGU journals varies greatly (Table S4): On average < 10% of all papers were highlighted in 2024 in most EGU journals. However, the proportion was nearly 30% in ESD whereas none were highlighted in GI and SE. In addition to the editorial highlighting, some EGU journals give a publication award for the most outstanding papers, which are selected by the executive editors or an independent committee (e.g., ACP Crutzen Publication Award; HESS Jim Dooge Award).

#### 4.1.3 Special issues and collections



Special issues (SIs) are compilations of articles about a specific topic within the scope of the journal, such as field campaigns, conferences or research themes. Unlike in traditional journals where SIs are published as separate (hardcopy) issues, SIs in EGU journals are fully virtual and articles are listed on a dedicated SI web page, in addition to publication on the regular

journal web page. SIs are typically open for submission for 1 to 2 years. This way, the publication of SI papers is not delayed by late submissions to the SI. All SI papers undergo the same process of peer review and publication as regular submissions.

Members of the scientific community can propose an SI in any EGU journal. Such requests can be made via a proposal specifying the SI's topic, duration and approximate number of papers, ideally accompanied by a preliminary list of paper titles. The executive editors of the journal check the proposal for its suitability for the journal and may request changes to the scope if needed. A large proportion of all SIs ( $\sim$ 230 of > 500 since 2001) is organized as inter-journal SIs, in which two or more journals participate (Section 4.3.1). These may include any EGU journal as well as other journals published by Copernicus.




In most EGU journals, SI submissions can be handled by dedicated SI guest editors in addition to regular editorial board members. Guest editors are usually nominated by the SI proposer but they require approval by the executive editors. In 2020, ACP introduced a change in the handling of SIs such that all submissions to SIs are handled by the regular ACP editorial board members (Section 4.1.4), and guest editors are no longer allowed. Two regular ACP editorial board members, who are not closely involved in the activities from which the special issue arises, act as SI coordinators. They oversee the SI in exchange with the executive editors but are not expected to handle the review process of all SI submissions. In addition, one to three SI co-organizers (often the proposers of the SI) are appointed to exchange with the authors and other members of the scientific community; they do not act as SI editors though.

The proportion of SI submissions, preprints and final journal papers in ACP clearly dropped during the last years from  $\sim$ 30% to  $\sim$ 10% (Figure 9). This trend may be partially ascribed to the new SI guidelines as introduced in 2020, but possibly also to the COVID-19 pandemic, during which time significantly fewer field studies and conferences took place that are often the scope of SIs. The percentages of preprints and final journal articles are similar, indicating comparable acceptance rates for regular and SI papers. The slightly higher proportion of SI preprints (relative to all preprints, i.e., regular and SI) compared to the proportion of submissions is likely due to the prior approval of the SI by the executive editors resulting in relatively fewer out-of-scope rejections.

Figure 9. Percentage of papers in ACP in special issues related to total ACP papers (2009 - 2024).

The organization and the overall trend in the proportion of SI papers in EGU journals are in contrast to those in journals by some large OA publishers that actively solicit an increasing portion of submissions via SIs that are typically handled by guest editors (often > 50% of all submissions) that frequently result in SIs that contain only very few papers (Brainard, 2023;

Petrou, 2023; Hanson et al., 2024). Publishers often advertise such special issues, both in terms of manuscript submissions and guest editorship; incentives for article submissions may even include the offer to publish articles at discounted rates. Such strategies, purely motivated to increase revenue, have led to a huge increase of special issues by large commercial publishers, often resulting in compilations with low scientific standards and little or no peer review. The resulting lack of scientific quality control in such SIs recently prompted the Swiss National Science Foundation to no longer fund any OA papers in special issues, in order to reduce the investment of public money on publications of low scientific quality and value (SNSF News, 2023). However, such measures seem inappropriate and unjustified for journals such as those of the EGU/Copernicus that apply and demonstrate consistent handling and transparent scientific quality control for all papers.

In 2024, the EGU journals introduced paper *collections* as an additional way to group papers on a specific topic. Similar to the SIs, collections can be organized across Copernicus journals as inter-journal collections, in which papers are (co-)listed on dedicated collection web pages, in addition to the regular journal pages. Unlike SIs that have a limited submission period (typically 1 - 2 years), collections do not have a pre-defined end date. Papers will be only included in collections upon their acceptance as final journal papers. This implies that all papers in collections have undergone the regular public peer review process handled by a regular editorial board member. The article selection for inclusion in a collection is made by the executive editors of the journal.

#### 815 4.1.4 Editor selection and duties





All editors appointed in EGU journals work on a purely voluntary basis without remuneration, in line with the not-for-profit philosophy of the EGU. Editorial boards are typically fairly large, so that each editor is expected to handle a comparably small number of manuscripts per year (e.g., 6 manuscripts per year in ACP). Editors are selected by the executive editors either upon nomination by colleagues or self-nomination via open calls or spontaneously. They are initially appointed for 3 years but they may stay for several terms, if they fulfill their duties over a longer period of time. In several EGU journals, outstanding dedication and performance of editors are recognized by annual awards, e.g., ACP Outstanding Editor Award; CP Editor Award. Each journal organizes editorial board meetings at least once a year and executive editor meetings as necessary. The executive editors of the EGU journals and the EGUsphere coordinator form the EGU Publications Committee, together with representatives of the EGU Executive Board and Copernicus Publications as ex officio members. Recently, a representative of the EGU Early Career Scientists (ECSs) was appointed as an additional member of the committee to allow for greater representation of early career perspectives and to enhance engagement with the ECS community.

All EGU journals apply consistent guidelines regarding EGU journal editor obligations. In addition, journals may develop specific guidelines, such as the ACP Editor Guidelines; GC Guidelines for Editors; GMD Editorial Policy and ACP author guidelines (Sections S1 and S2 in the Supplement) that are tailored to journal-specific topics and that may be frequently adjusted in response to questions or suggestions from the journal community and the global scientific community at large while following general scientific standards and codes of conduct.

The inaugural ACP executive editor committee in 2001 consisted of 5 members (U. Pöschl, T. Koop, K. Carslaw, B. Sturges, R. Sander), 3 of whom continued in this role for more than 20 years (U. Pöschl, T. Koop, K. Carslaw). In 2022, U. Pöschl and T.

Koop stepped down from their editorial duties and joined the ACP advisory board. To facilitate continuous and efficient oversight of the journal, it is currently led by two executive editors (B. Ervens, since 2019, and K. Carslaw). Instead of re-creating a larger executive committee, in 2021, the role of Senior Editors was introduced as an intermediate level between the  $\sim$ 160 regular ACP editors and the executive editors. Eight Senior Editors were selected by the executive editors as distinguished and experienced members of the editorial board based on their outstanding commitment and excellence as previously demonstrated in their editorial work. They were selected such that each journal subject area is covered by two Senior Editors according to their scientific expertise and core interests. The Senior Editors take responsibility for the editorial coordination of their subject area(s) and carry out a series of tasks to ensure scientific quality and foster submissions, in collaboration and exchange with the executive editors (Section S3). The appointment as an ACP Senior Editor is for two years, renewable upon mutual agreement with the executive editors.

Generally, all ACP editors can flexibly choose their workload throughout the year; they are expected to react promptly to the automated editor calls or manual assignments by Senior/Executive Editors (Section 3.1). In the initial years of ACP, authors of papers, which did not find an editor via automated calls in due time, were encouraged to contact an editorial board member of their choice to request the handling of their paper. If the authors were not successful, the paper was automatically rejected. Due to the considerable growth of ACP starting in  $\sim$ 2015, such personal handling requests led to imbalances in the workload of some editors. To facilitate a more even distribution of the workload and to avoid potential conflicts of interest, editors are now explicitly discouraged to respond to such requests. Instead, Executive/Senior Editors assign such papers to editors with low workload, or reject them if they are out of the journal scope or of low scientific quality. As editors often hesitate to handle papers of low quality or those at the edge of the journal scope, ACP introduced in 2020 the option of anonymous editor decisions at the access stage (before the preprint is accepted and posted for the public discussion; Section 3.2), i.e., allowing paper rejections on the behalf of the journal instead of individual editors. As soon as a preprint is accepted for the discussion stage, the editor's identity is automatically revealed to the authors.

#### 4.1.5 Referee selection








Peer reviewers have become the most limited resource in the entire scientific publication process due to the increase in the number of scientific submissions, publications and journals (Pöschl, 2012; Velterop, 2015; Willis, 2016; Severin and Chataway, 2021). The number of review requests per preprint in EGU journals has approximately doubled (from  $\sim$ 5 to  $\sim$ 10) since 2009 (Figure A2 b, d, f). To enhance the efficiency of referee identification and nomination, the publisher Copernicus continues to develop and improve search tools to suggest suitable referees (Copernicus News, 2023). Currently, editors can use the following tools that are provided via their editorial interface:

- The Copernicus' in-house development cREACTS ('Copernicus REferee ACTivity Score') ranks potential referees based on the likelihood that they provide a referee report, guided by the referee's statistics in Copernicus journals by considering accepted/declined/fulfilled referee calls during the preceding 12 months.

- The AI-based referee finder *Prophy* suggests up to 50 referees based on the semantic similarity in key expressions in the submitted manuscript to the history of publications or research proposals by the potential referee.
- The Copernicus' referee database lists previous referees according to their subject areas. All corresponding/contact authors of final accepted papers are automatically added to this database.
- Recent additions to the Copernicus referee database are listed separately as a subset of the full database. This list often includes ECSs, i.e., scientists within less than 7 years after their last degree.
  - Editors can make custom nominations based on their own preference.
  - Referees that are suggested by the authors, which is mandatory in most EGU journals.

This collection of referee identification tools provides editors with a wide range of referees at many career levels, including ECSs that represent more than half of the EGU membership. To broaden the pool of referees and to particularly increase the number of ECSs as peer reviewers, EGU organized hands-on peer review trainings in 2023 and 2024 for its members (Queiroz Alves and D'Souza, 2023), drawing on the extensive collection of preprints and referee reports as practical training resources. Upon successful completion, participants are added to the referee database and labeled as 'successful EGU peer review training participants'. As an additional training opportunity, a tandem review (co-review) scheme has been implemented in 2024 which allows reviewer teams, typically composed of an experienced referee who nominates a less experienced colleague to collaboratively prepare a referee report (Queiroz Alves, 2024), similar to the concepts in other journals, e.g., by Wiley/AGU (Dedej et al., 2023). In addition, interested scientists can apply to join the referee database via a form on the journal website, providing a CV and a brief statement on their publishing experience.

In several EGU journals, outstanding dedication and performance of referees are recognized by annual awards (e.g., *ACP Outstanding Referee Award*; *CP Referee Award*) in addition to the benefit of having their contribution documented by a citable report, for which they can also claim authorship by signing their comment in the public discussion (Sections 2.2 and 3.3).

#### 4.1.6 Processing times


The processing times for the six steps of the multistage peer review process, as detailed in Sections 3.1 - 3.6, are displayed in Figure 10 for the 19 EGU journals. The first panel shows the time between manuscript submission and its posting on EGUsphere and in the journal discussion forum. The first phase from the manuscript submission to initial decision (dark blue), includes the technical file validation by the Copernicus editorial support team and the assignment of the manuscript to a handling editor. This step takes less than two weeks on average for most journals. This time is similar for all journals, independent of whether they employ automated editor calls or use manual editor assignment, as it is done in the small journals with  $\lesssim 150$  submissions per year (ESurf, GChron, GC, NPG, SOIL, SE). The initial steps take the longest in the journal GMD (> 30 days). This may be due to the fact that its submission requirements are particularly strict as authors have to provide full codes of the underlying models that form the basis of their study (Sections 4.4 and S4.2); such checks and approval often require multiple rounds of file upload and validation. The total processing times, from submission to final paper publication,

are the longest for the 10 EGU journals that apply a post-discussion editor decision between the first and second stages of the review process (ANGEO, BG, CP, ESD, GC, GChron, HESS, NHESS, SOIL, TC) with on average 215 days vs 

**Figure 10.** Processing times [days] from a) submission to preprint publication; b) manuscript submission to final publication in the EGU journals (12-month median, December 2024).

# 4.1.7 Article processing charges






Article processing charges (APCs) are the fees that are charged for OA articles (Sections 2.1.2 and 2.1.3). They cover all costs for the production of peer-reviewed OA papers. They include the review support provided by the publisher office, online supplementary materials, typesetting, English language copy-editing, archiving and distribution of papers and interactive comments, i.e., the maintenance of websites and servers, electronic copies in open archives etc. The breakdown of APCs relative to total publications costs in journals (including those by the EGU) of Copernicus Publications is shown in Figure 11. The publisher's business profit margin (after taxes) is  $\sim$ 6%. This is significantly smaller than the margin of many commercial publishers that often exceeds  $\sim$ 30% (Van Noorden, 2013; Larivière et al., 2015; Pöschl, 2020; Butler et al., 2023). Copernicus Publications re-invests their surplus into training of new staff, journals owned and fully financed by the publisher, and enhancement of services and for outreach activities, operated by their non-profit association Copernicus e.V.

Until 2016, discussion papers (preprints) for EGU journals were typeset by the publisher, and authors were charged APCs upon publication of their discussion paper in a journal discussion forum. This procedure was changed in 2016, in analogy to other preprint servers and publishing platforms: No typesetting is performed and no APCs are charged at the first publication stage. Nevertheless, an initial technical file validation is applied free of charge, including a check of author names, affiliations, general formatting requirements etc. and the addition of the *EGUsphere* logo and citation to the pdf files. After successful peer review, APCs are charged for the main journal article. Supplemental information can be added free of charge as a separate file

(e.g., text, tables, figures), to which no copy-editing is applied by the editorial support. Additional assets can be connected free of charge to the paper, including video supplements or Jupyter notebooks.

Until the end of 2024, APCs were charged per page in most EGU journals (93 € or 77 € per page, using the publisher's packages for Word or LaTeX templates, respectively). GMD and CP started in 2021 charging fixed per-article APCs of 1600 € (net) per article; OS joined this pilot project in 2022. Since then, the article lengths in these three journals did not significantly change. As an investment, the EGU supports new journals during their start-up phase allowing for full waivers in the initial period and for reduced APCs during a transition time. APC discounts (-33%) were applied to all articles in GChron and WCD until the end of 2024; full waivers are still given to GC but full APCs are charged as of July 2025. Both GChron and WCD charge full APCs since January 2025. These three newest EGU journals, launched in 2018 and 2019 (Table A1) were accepted by Clarivate for the Web of Science in September 2023 and, thus, received their first JIF in 2024. This milestone in the journal development was used up to now as the criterion across the EGU journals to start charging APCs. An alternative criterion may be applied to journals launched in the future, given the limited relevance of the JIF (Section 2.4).





**Figure 11.** APC breakdown as an average of all journals published by Copernicus, based on the recommendations by the FAIR Open Access Alliance (FOAA, de Vries (2019)). Figure adapted from <a href="https://publications.copernicus.org/apc\_information.html">https://publications.copernicus.org/apc\_information.html</a> (last access: 24 Jan 2025).

Since January 2025, the APCs in all EGU journals are charged on a per-article basis to make the APC structure more transparent for all authors. The current APC scheme is structured into three journal groups with per-article net APCs of 1800€, 1350€ and 1980€, respectively (Table 3). Given that typesetting and copy-editing are major components in the paper production process (Figure 11), the journal groups were created based on the average article length using the statistics during the last three years. Short articles such as letters or peer-reviewed comments have approximately the same length in all EGU journals. Therefore, their standard net price is 900€ in all journals. As a benefit for its members, EGU, as the journal owner, grants a 10% APC discount for papers by corresponding authors (CAs) who are EGU members at the time of manuscript submission. This discount strengthens EGU's commitment to open science as a member-led, community-driven

learned society, distinguishing it from profit-driven interests by commercial publishers. As the largest European geoscientific society, EGU particularly aims at encouraging scientists from all regions in Europe to participate in the Union's activities, including publishing. Therefore, CAs affiliated in European economically disadvantaged (EED) countries<sup>6</sup> receive an automatic 50% APC discount on the standard or EGU membership APCs. CAs with affiliations in countries classified by *Research4Life* (Groups A & B) automatically receive a full APC waiver. These discounts/waivers likely further promote equality, diversity and inclusivity across the global geoscience community. By automatically applying them, the waivers/discounts reduce and hopefully diminish the geographical APC barriers of OA publishing as discussed by Klebel and Ross-Hellauer (2023).




In case authors affiliated in any country cannot afford the APCs in EGU journals, they can apply for a full or partial APC waiver during manuscript submission. Authors are asked to provide a brief justification for their request. Such discount and waiver requests are forwarded to the handling editor, who makes a recommendation, upon which an executive editor makes a final decision. Requests from any authors are considered, independently of their affiliation, unlike in many other journals, where waivers and discounts are reserved for authors from low/middle-income countries (Lawson, 2015). The total budget allocated for such waivers/discounts upon request, in addition to the automatic discount/waivers for EED and *Research4Life* is 10% of the previous year's total publication volume of all EGU journals; however, the actual amount spent on such discounts/waivers across all journals are usually well below this threshold. Links to the APC scheme and discount/waiver options are clearly displayed on all journal homepages through prominent buttons labeled as "Moderate Article Processing Charges" and "Financial Support".

**Table 3.** Net article processing charges (APCs) in EGU journals as of January 2025. Numbers in parentheses denote APCs after a 10% discount for corresponding authors who are EGU members; the same discount is applied as a basis for institutional agreements. Corresponding authors from European economically disadvantaged countries receive a 50% discount on APCs, while authors from countries classified under Groups A & B by *Research4Life* are eligible for a full waiver of APCs.

|                                                                                                                |                                                |               | APC (-10% for EGU members or institutional agreement) |               |  |  |  |  |
|----------------------------------------------------------------------------------------------------------------|------------------------------------------------|---------------|-------------------------------------------------------|---------------|--|--|--|--|
|                                                                                                                |                                                | Any country   | EED                                                   | Research4Life |  |  |  |  |
| Regul                                                                                                          | ar articles                                    |               |                                                       |               |  |  |  |  |
| I.                                                                                                             | ACP, AMT, BG, ESD, ESurf, GChron, HESS, NHESS, | 1800€ (1680€) | 900€ (810€)                                           | 0€            |  |  |  |  |
|                                                                                                                | OS, SE, SOIL, TC                               |               |                                                       |               |  |  |  |  |
| II.                                                                                                            | ANGEO, GC, GI, NPG, WCD                        | 1350€ (1215€) | 675€ (607.5€)                                         | 0€            |  |  |  |  |
| III.                                                                                                           | CP, GMD                                        | 1980€ (1782€) | 990€ (972€)                                           | 0€            |  |  |  |  |
| Short manuscripts: e.g., letters, peer-reviewed comments and some other journal-specific manuscript categories |                                                |               |                                                       |               |  |  |  |  |
| All EGU journals (I III.)                                                                                      |                                                | 900€ (810€)   | 450€ (405€)                                           | 0€            |  |  |  |  |

<sup>&</sup>lt;sup>6</sup> The classification is based on the list of 46 member states of the *Council of Europe* as well as Kosovo. EED countries are defined as European countries with a most recent gross national income per capita (GNI, World Bank) that is less than 25% of the maximum value of all European GNIs. The list is updated quarterly and currently includes Albania, Armenia, Azerbaijan, Bosnia and Herzegovina, Bulgaria, Croatia, Georgia, Greece, Hungary, Kosovo, Latvia, Lithuania, Moldova, Montenegro, North Macedonia, Poland, Romania, Serbia, Slovakia, Türkiye and Ukraine.

Copernicus Publications has signed numerous institutional agreements (IAs) with institutions, universities, funders and libraries for the centralized settlement of APCs. Unlike TAs, these IAs do not require giving access to subscription-based content since Copernicus has been a full OA publisher since 2004 (Section 2.1.3). Most of these IAs are based on a per-article basis, implying that the institution settles the invoice for the articles published by their authors in a given year. The underlying per-article APCs correspond to those as for EGU members in Table 3, i.e., applying a discount of 10% compared to the full prices. In the last years, a substantial proportion (24% in 2024) of APCs in EGU journals were settled via such IAs.

The business contract between the EGU as the journal owner and its service provider Copernicus Publications states that EGU receives the income from the APCs for all published papers in EGU journals after Copernicus covered the costs of its services (mostly working hours). The resulting difference between APC-generated income and the Copernicus expenditures is on the order of  $\sim$ 20 - 30%, which represents a significant source of income for the EGU as a non-profit learned society. This income is invested into activities of the Union (e.g., outreach, education, topical events and equality, diversity, inclusion), in accordance with the non-profit status of EGU. Thus, the EGU/Copernicus publication model is economically sustainable and able to finance a substantial part of the non-profit activities of a large scientific society such as the EGU, countering recent concerns that open access publishing may jeopardize the financial sustainability of such organizations (Brainard, 2025b).

The APCs in Table 3 are comparable to fees applied on publishing platforms, such as those managed by *F1000* (F1000 APCs; Open Research Europe (2020), Section 2.3.2). In comparison to journal APCs by many other publishers (Table S6), the APCs in the EGU journals are also much lower, independently of the sort of publication model and financing scheme (subscription-based closed access, hybrid or full OA, Section 2.1) (Schimmer et al., 2015; Pourret et al., 2021; Borrego, 2023).

# 4.2 The interactive community platform *EGUsphere*







### 4.2.1 Discussion papers, preprints, and conference contributions

Initially, manuscripts under peer review and discussion in an EGU interactive OA journal were termed 'discussion papers', such as in *Atmospheric Chemistry and Physics Discussion (ACPD)* that started in 2001 (Section 2.3.1). These papers were paginated in a regular volume, issue, page structure and had a DOI that included the journal and pagination. From 2016 onwards, discussion papers were no longer typeset, and had a DOI with the structure 'journalD-year-number' (with the D to reflect the discussion stage), disclosing that they had been under review for a journal. When such papers were rejected after being posted and discussed in a journal discussion forum, publishers outside the EGU journal family sometimes declined them for peer review in their journals since the DOI implied "previously rejected" or, falsely as "already published" journal papers. With the increasing popularity of preprints and publishing platforms for peer review of preprints across the scientific community, the term 'preprint' was adopted also for the EGU discussion papers as it more clearly indicates the status of a manuscript prior to potential publication in a journal (Section 2.3.1).

The interdisciplinary repository *EGUsphere* was launched in 2022, which merges the features of preprint servers with those of the established, well-functioning EGU journals with interactive multi-stage peer review. For the geosciences community, *EGUsphere* serves as a repository of non-peer-reviewed preprints and preprints that undergo peer review with the intention

of publication in one of the EGU journals, as well as conference abstracts and presentations. Unlike on traditional preprint servers, the complete peer review process of preprints intended for eventual journal publication is guided by a journal editor. Preprints that do not undergo peer review are not handled by journal editors but by preprint moderators. This differs from many other publishing platforms where editorial duties may only include the nomination of reviewers but no other editorial decisions. On such platforms, preprints may only receive reviewer ratings such as 'approved' or 'non-approved' without a final editor decision (Section 2.3.2).

All preprints on *EGUsphere* are assigned a DOI with the structure 'egusphere-year-number', regardless of whether they undergo peer review or not. Such a DOI clearly reflects that they are not (yet) peer reviewed papers but are preprints that may or may not undergo peer review. This change to the DOI setting, as compared to the initial format 'journalD-year-number' that included the journal name, eliminates previous problems faced by authors whose papers were rejected and subsequently declined submission to other journals. Only after successful revision and peer review completion, papers are published in the respective EGU journal with a DOI 'journal-year-firstpage'. During a transition period until the end of 2024, some EGU journals still offered both options, i.e., preprints were either posted on *EGUsphere* (DOI 'egusphere-year-number') and linked to the journal or only posted on the journal website with a DOI 'journalD-year-number'. As of January 2025, the latter route has been abandoned, i.e., all submissions to EGU journals are automatically labelled as an *EGUsphere* preprint. Journals indicate the journal relation during the open discussion on *EGUsphere* stating that the preprint is 'under review for journal'. This label is removed when a preprint reaches its final status (i.e., rejection or journal publication). Final journal publications are ultimately linked to the initial preprint.

Figure 12 displays the full concept of EGUsphere, including the three preprint options and conference contributions:

**Preprint Option I** corresponds to the former discussion papers that undergo the six steps of the interactive, public peer review process as described in Figure 3. Such preprints aimed at publication in an EGU journal undergo peer review, handled by a journal editor, according to the requirements and principles of the individual journals (discussion period, additional items during submission, etc). Also referees are expected to apply the standards of the target journal. At submission, authors have to choose *EGUsphere topics* in addition to journal subject areas. These keywords may be eventually used for an interdisciplinary, searchable index for the *EGUsphere* content. Preprints in the discussion phase are posted on the *EGUsphere* website and linked to the journal website to enhance their visibility. Upon publication of a paper in an EGU journal, the *EGUsphere* preprint and the full documentation of its public peer review and discussion are linked. Preprints submitted with the intention of journal publication that are, however, rejected at the access stage, may be recommended by the handling journal editor to become a stand-alone preprint (Option III). If authors refuse this option, there will not be any trace of the submission, neither on the journal website nor on *EGUsphere*.

**Preprint Option II** allows authors to seek peer review of preprints posted on other preprint servers that provide a DOI (e.g., *arXiv*). Such 'external preprints' maintain their original DOI while they undergo public discussion and peer review on *EGUsphere*, thereby avoiding double-preprinting, i.e., an assignment of two different DOIs to an essentially identical preprint. If authors would like to make changes to their original external preprint prior to the public discussion on *EGUsphere*, they are asked to create an updated version of their preprint on the external server. If their updates lead to a manuscript with similarity of

less than 60% as compared to the latest preprint version, their manuscript may be considered sufficiently different and treated according to Option I.

At any time during the discussion period, the authors can change their stand-alone preprint (Option III) into a preprint of Option I. Authors of stand-alone preprints are encouraged to actively solicit feedback from the community, which may help them to prepare a more elaborate manuscript for potential journal publication if they receive encouraging and constructive community comments on the technical soundness and relevance of the article. If authors choose to submit their manuscript for peer review to a journal outside the EGU family, eventually a link to the external journal article will be included on the page of the original *EGUsphere* preprint (Option III).

Currently, 98% of all submissions to EGUsphere are preprints aimed at journal publication (Option I, discussion papers), reflecting the high standing and popularity of the EGU journals. This preference is expected, because trends on other preprints servers show that  $\gtrsim 70\%$  of all preprints initially posted there are eventually published in a journal (Larivière et al., 2014; Abdill and Blekhman, 2019). While peer review of preprints is practiced on other publishing platforms with varying success, rigor and popularity (Section 2.3), EGUsphere takes advantage of the journal infrastructure, including experienced academic editors and a well-functioning, efficient workflow to ensure high scientific quality and the high reputation of the EGU journals in the geoscience community. Unlike other publishing platforms, EGUsphere combines the various preprint options and their straightforward conversion into each other, the documentation of their public peer review and discussion and the distillation of highlight articles in a single publishing framework. This structure avoids the need of overlay journals (Tennant et al., 2017) or external peer review platforms that lead to a disconnection of the preprints, their public discussion and peer review and final journal publication, which often results in low popularity and efficiency (Sandewall, 2012; Tennant et al., 2017).

In 2020, one part of *EGUsphere* was already launched as a repository for **conference material**, including conference abstracts and presentations for conferences organized by EGU, such as the annual *EGU General Assembly* (more than 20,000 participants in 2024). All conference abstracts submitted since 2015 have been retroactively included in *EGUsphere*. Before the conference, attendees can upload additional documents as supplemental material to their abstracts that can then be commented

Figure 12. Schematic of the interactive platform *EGUsphere* and its linkage to the EGU journals. The blue-shaded part indicates all options of the repository *EGUsphere*. The upper part (steps 1 - 6) corresponds to the journal workflow of EGU's interactive public peer review model (Figure 3, Section 3). External preprints (II) that are posted on preprint servers other than *EGUsphere* can undergo peer review on *EGUsphere* for consideration in one of the EGU journals. Stand-alone preprints (III) do not seek (immediate) peer review for an EGU journal; authors may decide to convert them into a preprint I at any time. Preprints (I) that are rejected from peer review by a journal editor may be posted as stand-alone preprint. Option IV denotes the collection of abstracts and presentation material for *EGU conferences*. Solid arrows denote mandatory actions, dashed arrows optional actions.

on by the conference participants until a limited period after the conference. The material is then permanently archived as a supplement to the conference abstract if the authors agreed on a CC BY or CC Zero license.

In summary, *EGUsphere* is an interdisciplinary repository for preprints and conference contributions in the geosciences. Its preprint repository hosts traditional preprints (Option III), but predominantly preprints with interactive discussion and peer review that are intended to lead to publication in an EGU journal (Options I and II).

# 4.2.2 Open science beyond OA and future perspectives for AI/ML







Going beyond other repositories and preprint servers, *EGUsphere* provides a seamless connection from preprints to journal articles and highlight selections as well as conference contributions. It offers functionalities for community-based commenting, discussion, and public peer review as well as other forms of open peer review and open science, including open data and open source as outlined below (Section 4.4).

Additional features from other open science initiatives and publishing platforms can be easily implemented. For example, informal community review can be introduced for stand-alone preprints that are not undergoing peer review in an interactive OA journal. Such preprints can be ranked in a similar way as, e.g., on publishing platforms managed by *F1000*, using labels such as approved/approved with reservations/not approved to reflect recommendations by community members engaging as reviewers without having been formally appointed by an editor. The outcome of such self-organized community review may encourage or discourage authors to pursue formal peer review in one of the EGU interactive OA journals or any other journal, which can build on the community ratings and comments already available on EGUsphere (overlay journals, Section 2.3). Beyond that, *EGUsphere* offers opportunities to explore the benefits and shortcomings of new technologies for scientific review and quality assurance, in particular related to the rapid recent evolution of artificial intelligence and machine learning (AI/ML) tools. EGU recently published a statement on the use of AI-based tools for the presentation and publication of research results in Earth, planetary and space science (EGU News, 2024a). Concerns have been raised that AI-generated papers may undermine the integrity of scientific publishing by introducing content that lacks originality and quality (Leung et al., 2023; Májovský et al., 2023; Resnik and Hosseini, 2024). In this regard, the publication of preprints and their public peer review on *EGUsphere* enables the identification and disclosure of potential misuse of AI-generated content in the scientific discourse, including manuscripts and review reports.

The interactive OA publishing approach (publish-then-review) effectively reduces ethical and legal concerns regarding the upload of unpublished material into AI/ML tools such as large language models (LLMs) for reviewing purposes (Resnik and Hosseini, 2024) because preprints posted on *EGUsphere* are freely available for re-use under the CC BY license. AI tools may be applied to produce summary reports aimed to identify deficiencies of manuscripts such as unclear or incomplete descriptions of methodologies, results, and conclusions (Zhuang et al., 2025). Such reports, clearly labeled as AI-generated, could accompany preprints on *EGUsphere* to provide optional input for community and peer review, whereby community members and referees can decide to use or discard the AI-generated information. If properly applied, such use of AI may benefit and enhance the efficiency and rigor of scientific quality assurance (Hosseini et al., 2025). Note, however, that the

involvement of AI in the peer review process cannot replace human referees because human scientists must remain the actual "peers" of human authors.

# 4.3 Interdisciplinary exchange and virtual compilations








As outlined in the previous section, *EGUsphere* in connection with the EGU journals provides a structure for a wide spectrum of non-peer-reviewed and peer-reviewed scientific articles in terms of manuscript types and geoscientific topics. The consistent handling of manuscripts within the 19 journals therefore allows the creation of interdisciplinary virtual compilations that group articles from different journals. Such grouping can take place by topic in form of inter-journal special issues or collections (Section 4.1.3) or by manuscript type, such as review articles or letters. The extension of the two-stage publication process to virtual compilations takes advantage of the established journal infrastructure and does not require the creation of new journals that may compete with the existing ones for the best articles. Since all editorial decisions are published as part of the documentation of the public peer review in the individual journals, each article in the compilations has undergone the same rigorous scientific quality assurance as any other article published in EGU journals.

Figure 13 shows how the four virtual compilations that emerged from the EGU journals are connected to the two-stage publication process. It also illustrates the level and intensity of quality assurance ("distillation") that is applied to the papers in each of these compilations: Whereas articles in inter-journal special issues are published upon the approval by the handling editor only, the inclusion of review articles and letters in the *Encyclopedia of Geosciences* or the *EGU Letters*, respectively, requires an additional executive editor decision step.

# 4.3.1 Inter-journal special issues and collections

Inter-journal special issues (SIs) are organized across two or more journals on topics that are within the scope of these journals. Any EGU journal but also other journals by Copernicus Publications may be part of an inter-journal SI. The handling of SI manuscripts in each of the journals is determined by the journal-specific SI guidelines (Section 4.1.3). Averaged over all EGU journals, nearly half (46%) of all special issues are inter-journal SIs whereas in some EGU journals (e.g., ACP, AMT), they represent the majority of special issues (Figure 14a and Table S5). About 60% of all inter-journal SIs are organized between 2 journals, but the number of journals participating in an inter-journal SI has been (so far) as high as 6 (Figure 14b). All papers, i.e., preprints and final journal papers, of inter-journal SIs are listed on the websites of every journal that participate in the SI thereby enhancing the visibility and interdisciplinary character. The strong journal overlap in special issues is illustrated in Figure 14c. A similar concept of inter-journal special issues or special collections was introduced in AGU journals in 2012 (Hanson and van der Hilst, 2017), i.e. several years after the EGU journals started this concept.

Some journal combinations, e.g., ACP and AMT, are quite frequent due to the complementary and overlapping journal scopes: 95% of all inter-journal SIs in AMT (72 of 76) are co-organized with ACP. Since ACP has organized more inter-journal SIs in total (104), these SIs correspond to 69% of all inter-journal SIs in ACP. The wide variety of journal combinations demonstrates that the inter-journal SIs provide an efficient and popular way to create interdisciplinary compilations, in which all articles undergo the same transparent peer review process. The linking of papers to SIs across journals is not only applied

**Figure 13.** Extension of the the two-stage publication process in EGU journals (Section 3) by a third step to populate the virtual compilations. Details on inter-journal special issues and collections, the *Encyclopedia of Geosciences* and *EGU Letters* are given in Sections 4.3.1, 4.3.2 and 4.3.3, respectively.

to EGU journals but extends also to some other Copernicus journals, including *Aerosol Research (AR)*, *Advances in Statistical Climatology, Meteorology and Oceanography (ASCMO)*, *Earth System Science Data (ESSD)* and *History of Geo- and Space Sciences (HGSS)*. The particular role of ESSD (launched in 2009) is evident since it has inter-journal SIs with nearly all EGU journals. ESSD focuses on publication of comprehensive data documentation, including their sources, codes and algorithms (Carlson and Oda (2018) and Section 4.4) while the interpretation or discussion of their implications are included in the articles of the discipline-specific EGU journals.



SIs allow submissions only during a limited time period (typically 1 to 2 years). The inclusion to a particular SI is performed upon request by the authors during manuscript submission. As of 2025, theme collections can be organized on specific topics with the scopes of one or more journals, in addition to special issues. Such collections are also organized as inter-journal compilations and are open without defined end date to group papers on specific topics. The linking of papers to collections is performed upon final paper acceptance; thus, no preprints are included in collections.

**Figure 14.** a) Fraction of total single-journals and inter-journal special issues; b) number of journals involved in inter-journal SIs for each EGU journal; c) portion [%] of inter-journal SIs in the journals in the columns (top axis) paired with journals in the rows (left axis). The color-coded pattern of rectangles is not symmetric as the relative proportions [%] are shown and different journals have different absolute numbers of inter-journal special issues. In addition to EGU journals (Table A1), other Copernicus OA journals are included: *Aerosol Research* (AR), Advances in Statistical Climatology, Meteorology and Oceanography (ASCMO), Earth System Science Data (ESSD) and History of Geo- and Space Sciences (HGSS).

## 4.3.2 Encyclopedia of Geosciences - A collection of scientific review articles

The *Encyclopedia of Geosciences* was established in 2017 and comprises 507 review articles (as of December 2024) that are published in one of the 19 EGU or other Copernicus journals since 2001 (Figure 15). All 19 EGU journals consider the manuscript type review articles (Table S1); review articles in EGU journals are expected to summarize the status of knowledge on a particular topic and outline future directions of research within the scope of the journal. Occasionally also other manuscript types may be considered for inclusion in the *Encyclopedia of Geosciences* if they contain a sufficiently broad overview of a relevant topic (e.g., some ACP Opinion articles and NHESS Invited Perspective articles). Articles that fulfill these criteria are added to the *Encyclopedia of Geosciences* after their public peer review and acceptance for publication according to the criteria in the given EGU journal (Section 3). During or after the peer review process, authors interested in having their review article

included in the Encyclopedia are requested to contact the editors of the Encyclopedia. Alternatively, the handling journal editor or possibly a referee can also make the recommendation to the editors of the Encyclopedia.

The editorial board of the *Encyclopedia of Geosciences* is composed of one editor from each contributing journal. Articles in the Encyclopedia keep the DOI of their original journal article. This way, they are linked to the full documentation of their review process and also benefit from the journal community and reputation. To emphasize its interdisciplinary encyclopedia character, searches can be performed based on 11 *EG topics* and 183 *EG index terms*. In addition, readers can also find articles based on author names, title or words in the abstract, just like for any EGU journal.

**Figure 15.** Top: Number of articles added to the Encyclopedia of Geosciences per year (blue) and total articles included (black), bottom: number of articles added per journal and year.

### 4.3.3 EGU Letters - The EGU highlight magazine

More than a decade ago, Pöschl (2012) laid out the potential of EGU's multi-stage, public peer review concept for a third stage to create interdisciplinary compilations of specific journal articles of highest quality and particular significance, as opposed to (potentially) high JIF journals where such selection criteria are often hidden. *EGU Letters*, launched in 2023, is a compilation of letter-style articles that underwent peer review in one of the 8 EGU journals (ACP, BG, ESurf, ESD, GC, NPG, OS, SOIL) that currently consider the manuscript category Letters (Section 4.1.2). Letters are concise, engaging articles (≤ 2500 words) reporting exceptionally important results and significant scientific advances in geoscientific research that are of high general

interest to the entire geoscientific community and/or to the broader public (and media). Letters include both of the following characteristic features:

- Important discoveries and research highlights in geoscientific research.
- Solutions to or progress with long-standing and important questions in their research area.

At submission, authors select the manuscript type *Letter* and have to explain how their manuscript meets these criteria. During the review process, Letters are expected to be rated as outstanding/excellent in the three principal review criteria, i.e. scientific significance, scientific quality and presentation quality (Section 3.2). If not accepted as a letter, the manuscript may still be accepted for final publication as a regular research article and possibly be ranked as a highlight article ('editor's choice', Figure 5). The decision on the highlight and letter status is made by the journal executive editors, upon recommendation by the handling editor. The editorial board of *EGU Letters* is informed of each letter acceptance in a journal. This board is coordinated by a chair person and is comprised of the executive editors of the eight participating EGU journals that currently consider the manuscript type letters. Upon favorable decision by the *EGU Letters* editorial board, the paper is displayed on the *EGU Letters* website, thus, providing an even higher visibility to a broader (geoscience) community than being published on the individual journal website only. Articles included in *EGU Letters* keep the DOI of their original journal publication.

**Figure 16.** Evolution of *EGU Letters* since its launch in 2020. Colored bars show the number of letters from individual journals per year; the black line is the total number of *EGU Letters* (2020 - 2024).

The first letter was published in 2020; as of December 2024, 34 journal letters are included in *EGU Letters* that are published in ACP, BG, ESD, GC, NPG, OS or SOIL, respectively (Figure 16). The first ESurf letter was pblished in May 2025. The 11 letters published in 2023 correspond to about 0.4% of all EGU publications and about 1% of papers published in the 5 journals that published letters in that year whereas these numbers are 0.3% and 0.8% in 2024, respectively, indicating that these articles represent a very exclusive fraction of all published papers. Implementing this third step as an upward process upon the publication of papers in one of the EGU journals allows the efficient selection of outstanding articles of particular interest to the full geoscience community, without compromising scientific completeness and quality assurance. After their

transparent three-tier selection process, *EGU Letters* are expected to be of comparable quality and higher credibility than articles of similar formats in other high-impact, interdisciplinary journals.

# 4.4 Interactive OA publishing and EGU/Copernicus in the global landscape of open science






Interactive OA publishing with public peer review and discussion as practiced in the journals, virtual compilations and community platform of EGU/Copernicus integrates multiple aspects and forms of open science, including open access, open data, open source, and open peer review to enhance the accessibility, traceability, and quality assurance of scientific knowledge. As widely acknowledged by national and international declarations (e.g., Berlin Declaration on Open Access to Knowledge in the Sciences and Humanities (2003); Science Europe Strategy Plan 2021-2026 (2021); Second French Plan for Open Science (2021); SPARC Europe 2025-2028 (2025)) and in the *Data Policy of EGU/Copernicus*, the output of research consists not only of journal articles but also of data sets and model code etc.

Only a comprehensive account of relevant information can guarantee integrity, transparency, re-use and reproducibility of scientific findings. Moreover, all of these resources provide great additional value in their own right. The *Declaration of Research Assessment* (DORA, 2012) and the *Coalition of Advancing Research Assessment* (CoARA, 2022), signed by EGU in 2024 (EGU News, 2024b), recognize the importance of this wide array of research output for scientific quality and impact. It is desirable that data, code and other information underpinning the research findings are "findable, accessible, interoperable, and reusable" (FAIR Principles (2016), Wilkinson et al. (2016)) not only for humans but also for machines.

As a signatory of the Commitment statement by the Coalition on Publishing Data in the Earth and Space Sciences (COPDESS) (2014) and the Enabling FAIR data Commitment Statement in the Earth, Space, and Environmental Sciences (2018), Copernicus Publications adheres to such best practices in sharing open data to advance reproducibility, transparency and innovation in Earth and space sciences. Their practices follow the recommendations by the Future of Research Communication and e-Scholarship (FORCE11, 2011) initiative regarding Data Citation Principles (Martone, 2014) and Software Citation Principles (Smith et al., 2016) through the effective use of information technology, with an emphasis on improving knowledge creation and sharing in digital publications.

The *Data Policy of EGU/Copernicus* (Section S4.1) requests depositing data that correspond to journal articles in reliable (public) data repositories, assigning digital object identifiers, and properly citing data sets as individual contributions. In addition, data sets, software, algorithms, model code, video supplements, video abstracts, International Geo Sample Numbers and other digital material should be linked to the article through DOIs. Authors are requested to list citations to the data and assets in a data availability statement at the end of each article. If data are not publicly accessible at the time of publication, the data statement must specify when and where they will become available and how readers can access them until then. Authors should provide embargoed data to referees during review to ensure reproducibility, labeled as "reviewer-only access" which implies that referees agree not to copy, share or reuse the data. If public deposition is not possible (e.g., due to commercial constraints), a detailed justification must be provided.

To meet the needs and preferences of the scientific communities served by the EGU/Copernicus journals, the data policy can be flexibly adapted to align with requirements and practices in different fields, while ensuring transparency, openness,

accessibility and re-usability for all as far as practicable. This is achieved by journals that are solely focused on the publication of datasets or codes or by individual requirements within individual journals. Examples include:

- Geoscientific Model Development (GMD) is an EGU/Copernicus journal dedicated to the publication of the description, development, and evaluation of numerical models of the Earth system and its components. Its journal data policy requests precise versions of all code and data associated with the paper to be deposited in persistent public archives (Section S4.2).
   Information on access to other versions of the code and data as well as the licence of the code, according to Open Source Definitions should also be provided.
- Earth System Science Data (ESSD), a Copernicus journal, publishes data papers, including peer review of original research data (sets) to foster the re-use of high-quality data through easy, free, and open exchange of high quality datasets in the Earth sciences (Carlson and Oda, 2018). An ESSD "product" consists of a detailed description published in ESSD, linked to a dataset archived in a reliable data repository with a permanent identifier (Section S4.3). ESSD is efficiently linked to publications in EGU/Copernicus OA journals or on EGUsphere, e.g., through inter-journal special issues (Section 4.1.3, Figure 14).
- Atmospheric Chemistry and Physics (ACP) offers various manuscript types (Section 4.1.2, Table S1) that have different
   requirements in terms of data reporting. Measurement Reports strictly request that all underlying data are made openly accessible and citable with a functional DOI. In other manuscript types exceptions from the data policy can be made, if authors convincingly justify why data cannot be publicly shared.

Generally, it is strongly recommended for all papers though to share data needed to replicate figures in the published articles, ideally in a public repository or, alternatively, in a supplement to the paper. These practices and policies contribute to the overall goal of open science by promoting reproducibility, accessibility and scientific quality assurance to ultimately contribute to the epistemic web of knowledge through transparent knowledge sharing. At the same time, EGU and Copernicus respect the preferences of different scientific communities, where the disclosure and re-use of data and source code may progress under different conditions, at different levels, and at different rates.

### 5 Summary and conclusions




Over the past decades, individual scholars, scholarly organizations, learned societies and commercial publishers have experimented with new models of scholarly publishing, including open access and hybrid journals, open repositories and publishing platforms, and various forms of open peer review and other elements of open science. The interactive OA publishing approach of the European Geosciences Union (EGU)/Copernicus was among the first and most successful initiatives combining open access and public open peer review with community discussions in top quality scientific journals. Since 2001, more than 50 000 journal articles and 60 000 preprints/discussion papers accompanied by more than 250 000 comments by reviewers, authors, editors and the broader scientific community have been published in the 19 interactive open access journals by the European Geosciences Union (EGU) and Copernicus Publications. Today, the EGU publication portfolio extends beyond the

EGU journals and includes the preprint repository EGUsphere and virtual compilations of published journal articles, including the Encyclopedia of Geosciences for review articles and EGU Letters for concise highlight articles.








The average costs and charges of article processing were below 1000 € per article at the start of ACP in 2001 and following years. Currently, the article processing charges (APC) of EGU/Copernicus are still below 2000 € per article. Moreover, 10% of the total publication volume are available for optional APC discounts and waivers in addition to automatic waivers for papers from countries classified by *Research4Life*. The difference between APC-generated income and publisher expenditures is on the order of ~20 - 30%, which represents a significant source of income for the EGU as a non-profit learned society. This income is invested into activities of the Union (e.g., outreach, education, topical events and equality, diversity, inclusion), in accordance with the non-profit status of the EGU. Thus, the interactive OA publishing model of EGU/Copernicus is economically sustainable and able to finance a substantial part of the non-profit activities of a large scientific society such as the EGU. Overall, the interactive OA publishing approach developed and practiced by ACP, EGU, and Copernicus results in a unique combination of achievements, which has not been reported for any other scientific publishing approach: top scientific quality and visibility/impact in combination with low rejection rates, moderate costs, and long-term financial sustainability. Twenty-five years of experience with interactive OA publishing by the EGU demonstrate that learned societies can financially benefit from OA publishing in similar ways as from subscription publishing, but generate much more benefit for science and humanity overall.

In spite of a growing number and volume of successful OA journals and publishing platforms - including those of EGU/ Copernicus, PLOS, BioMed Central, SciPost and other scholarly initiatives and commercial publishers - the overall transition from traditional subscription journal publishing to OA proceeded initially very slowly. Although hundreds of leading scholarly organizations around the world committed to OA by signing the Berlin Declaration on Open Access to Knowledge in the Sciences and Humanities (2003), only about 10% of all articles among the large number of scientific journal articles (> 2 million per year, published in > 20,000 peer-reviewed journals) were OA in 2013. This growth rate of 1% per year would have implied many more decades to achieve OA to a majority of scholarly research publications. This slow movement was largely due to a preference of researchers to publish in traditional journals, and to the reluctance of large traditional publishers to change their highly profitable subscription business with profit margins of 30% and more, and to offer OA without high extra charges (thousands of Euros per OA article), in addition to the already excessively high subscription income (approx. 4000€ per journal article). To accelerate this progress, leading scholarly organizations united in the global initiative *OA2020* to replace traditional subscription contracts by transformative agreements. These agreements provide cost-neutral or even costsaving ways to establish OA to articles from authors of the participating institutions/consortia, while maintaining access to subscription-based content from other institutions/consortia that have not yet established OA to articles from their authors. Transformative agreements boosted the proportion of OA to articles from some institutions and countries to 90% and more. Nevertheless, and beyond the transformation from subscription to open access, it remains crucially important to uphold and promote further improvements in scientific publishing by innovative OA publishers.

Thus, we propose to include the following measures in the ongoing and future development of open access publishing and open science:

• Complement transformative OA agreements with traditional publishers by equivalent OA publishing agreements with new and innovative OA publishers and platforms. Community-based, non-profit/not-for-profit initiatives have proven to drive innovation and are intrinsically motivated to improve scholarly development rather than just maximizing economic profits like some (semi-)predatory commercial publishers. As learned societies are generally motivated by scientific quality rather than by commercial interests, they should be enabled and encouraged to offer and sustain high-quality publishing platforms and journals, with financial support via proper OA publishing agreements that may be analogous to or build on well-established and functioning elements of existing transformative OA publishing agreements.





- Introduce appropriate elements of transparency and open peer review in all peer-reviewed OA publications to counteract (semi-)predatory publishing and deterioration in scientific quality assurance that undermine scientific quality and reliability. This can be done in form of public peer review and interactive discussion as practiced by EGU/Copernicus, or by partial/full disclosure of the pre-publication history (review reports) upon publication of the accepted paper. Such transparency provides valuable insights into the scientific discourse and evolution of scientific ideas and knowledge.

  EGU/Copernicus journals have been publishing reviewer reports for over 25 years, and more recently other journals, such as Nature, have also made this practice mandatory (Nature Editorial, 2025), which are major steps towards an epistemic web (Figure 1). Public reviewer and community comments should be appreciated and counted as valuable major contributions to the scientific discourse, e.g., in scientific assessments and evaluations related to recruitment, promotions, grants and honors/awards.
  - Utilize interactive OA publishing platforms to explore the potential of alternative forms and new technologies for scientific review and quality assurance. Such features include different types of reviews and ratings from community members, reviewers, moderators, and editors. The scientific community should be encouraged to take advantage of such platforms or other outlets for early sharing and discussion of their ideas and results.
  - Test and utilize AI/ML tools to support and complement human reviewers in the process of scientific review and quality assurance. For example, we suggest to include reports, clearly labeled as AI generated, that may aid in identifying deficiencies of specific aspects or sections of a manuscript (methodologies, references etc.). Such AI-generated reports, however, should not replace the assessment by human peers (proper peer review). The benefits, risks and development of such tools should be explored and monitored following clear guidelines and standards of scientific integrity and ethics.
  - Integrate OA with other forms of open science such as open data and open source according to the *FAIR Principles* ('findable, accessible, interoperable, and reusable'). The rate and extent of advancing these principles in various fields can be adjusted according to the needs and preferences of different disciplines and communities.

Overall, the interactive OA publishing approach has greatly advanced open science and scientific quality assurance. It provides a basis to develop an epistemic web of knowledge that displays the scholarly discourse by showing what we know, how well we know it, and where the limitations are.

**Table A1.** OA publications of the European Geosciences Union (EGU). Upper part: Journals and the year when they implemented the multistage interactive publishing model. Lower part: Additional OA EGU publications

| EGU open-access journals                   |                                | Acronym                                                        | Multistage open                               | Websi                                                     | te https://w                 | vww. <sup>6</sup>                 |  |
|--------------------------------------------|--------------------------------|----------------------------------------------------------------|-----------------------------------------------|-----------------------------------------------------------|------------------------------|-----------------------------------|--|
|                                            |                                |                                                                | peer review since                             |                                                           |                              |                                   |  |
| Annales Geophysicae                        |                                | ANGEO 2018 <sup>1</sup>                                        |                                               | annales-geophysicae.net                                   |                              |                                   |  |
| Atmospheric Chemistry and Physics          |                                | ACP 2001                                                       |                                               | atmospheric-chemistry-and-physics.net                     |                              |                                   |  |
| Atmospheric Measurement Techniques         |                                | AMT 2008                                                       |                                               | atmospheric-measurement-techniques.net                    |                              |                                   |  |
| Biogeosciences                             |                                | BG 2004                                                        |                                               | biogeosciences.net                                        |                              |                                   |  |
| Climate of the Past                        |                                | CP 2005                                                        |                                               | climate-of-the-past.net                                   |                              |                                   |  |
| Earth Surface Dynamics                     |                                | ESurf                                                          | 2013                                          | earth-surface-dynamics.net                                |                              | namics.net                        |  |
| Earth System Dynamics                      |                                | ESD                                                            | 2010                                          | earth-s                                                   | earth-system-dynamics.net    |                                   |  |
| Geochronology                              |                                | GChron                                                         | 2019                                          | geochronology.net                                         |                              |                                   |  |
| Geoscience Communication                   |                                | GC                                                             | 2018                                          | geosci                                                    | geoscience-communication.net |                                   |  |
| Geoscientific Instrumentation, Methods and |                                | GI                                                             | 2011                                          | geoscientific-instrumentation-methods-and-data-systems.ne |                              |                                   |  |
| Data Systems                               |                                |                                                                |                                               |                                                           |                              |                                   |  |
| Geoscientific Model Development            |                                | GMD 2008                                                       |                                               | geoscientific-model-development.net                       |                              |                                   |  |
| Hydrology and Earth System Sciences        |                                | HESS 2004 <sup>2</sup> hydrology-and-earth-system-sciences.net |                                               | arth-system-sciences.net                                  |                              |                                   |  |
| Natural Hazards and Earth System           | NHESS                          | $2013^{3}$                                                     | natural-hazards-and-earth-system-sciences.net |                                                           |                              |                                   |  |
| Nonlinear Processes in Geophysics          | NPG                            | 2014 4                                                         | nonlinear-processes-in-geophysics.net         |                                                           |                              |                                   |  |
| Ocean Science                              |                                | OS                                                             | 2005                                          | ocean-science.net                                         |                              |                                   |  |
| Solid Earth                                |                                | SE                                                             | 2009                                          | solid-e                                                   | solid-earth.net              |                                   |  |
| SOIL                                       |                                | SOIL                                                           | 2014                                          | soil-jo                                                   | soil-journal.net             |                                   |  |
| The Cryosphere                             |                                | TC                                                             | 2007                                          | the-cry                                                   | the-cryosphere.net           |                                   |  |
| Weather and Climate Dynamics               |                                | WCD                                                            | 2019                                          | weath                                                     | weather-climate-dynamics.net |                                   |  |
| Other EGU publications                     | Acronym                        | Description                                                    |                                               |                                                           | Launch                       | Website https://www. <sup>6</sup> |  |
| Advances in Geosciences                    | ADGEO                          | Proceedings                                                    |                                               |                                                           | 2003                         | advances-in-geosciences.net       |  |
| Encyclopedia of Geosciences EG             |                                | Virtual compi                                                  | lation of review artic                        | eles                                                      | 2017                         | encyclopedia-of-geosciences.net   |  |
| EGU Letters                                | Virtual EGU highlight magazine |                                                                |                                               | 2020                                                      | egu-letters.net              |                                   |  |
| EGUsphere                                  | Preprint and o                 | community platform                                             |                                               | 2022 5                                                    | egusphere.net                |                                   |  |

<sup>&</sup>lt;sup>1</sup>ANGEO was launched in 1983 by the European Geophysical Society; between 1994 and 2000 (volume 19), it was published by Springer; since then (volume 19) it is an EGU/Copernicus journal that became OA in 2009.

<sup>&</sup>lt;sup>2</sup> HESS was launched in 1997, became OA in 2004.

<sup>&</sup>lt;sup>3</sup> NHESS was launched in 2001, became OA in 2004.

<sup>&</sup>lt;sup>4</sup> NPG was launched in 1994, became OA in 2004.

<sup>&</sup>lt;sup>5</sup> EGUsphere was launched as a repository for conference abstracts in 2020; conference contributions since 2015 were retroactively included.

<sup>&</sup>lt;sup>6</sup> All links were last accessed on 24 Jan 2025.

**Figure A1.** Rejection rates in the EGU journals (2009 - 2024). Top and middle panels: Individual journals; bottom panels: Sum of all EGU journals. a, c, e) at the access stage, prior to public discussion and open peer review; b, d, f) after discussion and peer review. The grey symbols in f) show the average rejection rates for journals with *vs* without editor decision immediately after the discussion (Section 3.4)

**Figure A2.** Overview of requested and received peer review reports in EGU journals (2009 - 2024). Top and middle panels: Individual journals; bottom panels: Sum of all EGU journals. a, c, e) Number of peer review reports per preprint, b, d, f) Number of peer review requests per preprint.

Data availability. All data are either publicly available on the websites of the journals (https://www.egu.eu/publications/open-access-journals/) or they were kindly provided by the publisher Copernicus (https://publications.copernicus.org/). Data for Figures 4, 6 - 10, 14, A1 and A2 are available at https://zenodo.org/records/14713159, Ervens (2025).

Author contributions. BE and UP developed the idea of the paper and wrote the manuscript. KC and TK contributed by means of critical review and commentaries to all versions of the manuscript.

Competing interests. Two authors are members of the ACP editorial board; two authors are members of the ACP advisory board.


Acknowledgements. All current and former members of the ACP advisory and editorial boards and the previous ACP executive editors Bill Sturges, Rolf Sander, Maria Cristina Facchini are gratefully acknowledged. In addition, we thank Martin Rasmussen, Natascha Töpfer and the complete Copernicus team for the successful, enjoyable and productive collaboration to develop and grow the EGU publications. We appreciate support and helpful comments by Eduardo Queiroz Alves and the other members of the EGU office. In addition, we would like to thank all referees, authors and the entire EGU journal community for their continuing support and contributions to the interactive open access EGU journals. We thank Ludo Waltman, Mingjin Tang and an anonymous referee for their valuable comments during the public discussion of our paper. We are also grateful to Kai Geschuhn (Max Planck Digital Library) and Nina Schönfelder (Universitätsbibliothek Bielefeld) for their constructive feedback.

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
