# Peer review of "Review of interactive open access publishing with community-based open peer review for improved scientific discourse and quality assurance"

_EGUsphere, 2025_

## Author Response (AR1)

**Author response**

We thank both referees and Mingjin Tang for their thoughtful comment. We respond to them below; referee comments are shown in black, our author responses in blue; suggested new manuscript text is indicated in red.

In addition to these changes, we made some other improvements throughout the manuscript which are highlighted in the track change version of the paper. They include

- We realized that our paper may be better suited for the manuscript category 'Review articles' rather than 'Opinions' in ACP. Adjusting the manuscript title and category took care of some of the concerns that had been raised about the length of the paper, by the referees, the editor and other colleagues.
- We added Table S6 as additional information to illustrate the moderate article processing charges of EGU/Copernicus journals in comparison to (geo)scientific journals by other publishers.
- We added some references that came to our attention since we submitted the first version of the paper.

**Referee #1:**

This paper provides a comprehensive overview of the publication process of the EGU journals, with particular focus on the value and practice of open access and open review. The paper is a valuable documentation of the evolution of open access journals and engagement of authors, reviewer, editors and the general scientific community. I want to commend what the leadership teams of the EGU journals have done and congratulate them for the accomplishment in a relatively "short" period of 25 years. I only have a few minor comments listed below.

1. Figure 10 on page 34, the processing time between the acceptance date to publication date appears to be very long. I wonder why this is so.

Author response: The time from acceptance to publication is determined by the production process on the publisher side including the turn-over time of final revised files and manuscript proofs with the authors. Where necessary this may also involve additional iteration with the editors, e.g., in case of substantial changes during proofreading which is a quality assurance feature which may be less valued/applied by other journals and publishers. This time used to be shorter and has increased during recent years. The publisher is trying to increase the working capacities and further streamline the production process.

2. It will be valuable to the readers if the authors can provide their perspectives on how the publication process and quality of the journals can be further improved. While the final section contains some discussions, they are mostly about the technical aspects. Readers will interested to hear about existing challenges and broader perspectives.

**Author response:** We thank the referee for this suggestion to expand the Summary and Conclusion section to emphasize that our suggestions are not limited to technical features but also involve the scientific community by actively recognizing and implementing features of open science.

Learned societies play a crucial role as they are generally motivated by scientific quality rather than by commercial interests. Societies should be encouraged to provide publishing platforms/journals, with financial support/incentives via OA publishing agreements. Examples include initiatives by the German Research Foundation or the German National Academy of Sciences *Leopoldina*). This can be greatly advanced and promoted when/if research funding and performing organizations support innovative high quality OA publishers like Copernicus through efficient and modular OA publishing agreements, comparable to national and international transformative agreements with traditional publishers (e.g., DEAL, OA2020). Accordingly, we added to the first bullet point in Section 5 (Summary and conclusions):

As learned societies are generally motivated by scientific quality rather than by commercial interests, they should be enabled and encouraged to offer and sustain high-quality publishing platforms and journals, with financial support via proper OA publishing agreements that may be analogous to or build on well-established and functioning elements of existing transformative OA publishing agreements.

Higher acceptance and recognition of public review comments by the scientific community does not only require technical improvements by publishers. We also suggest that public commenting should become more valued, common place and a standard in scientific publishing, and that such reviewer and community comments should be valued as a major contribution to the scientific discourse, e.g., in scientific assessments and evaluations related to grants, recruitment and promotions. This is also in line with the suggestions by CoARA that identify peer review as being central and the most robust method known for assessing scientific quality by the research community.

To emphasize that this should not be only considered a technical feature but also driven by the scientific community, we extended the second bullet point in the section by

Public reviewer and community comments should be appreciated as valuable major contributions to the scientific discourse, e.g., in scientific assessments and evaluations related to recruitment, promotions, grants and awards.

In line with the suggestions by CoARA, also the scientific community should explore and value different forms of scientific output. This includes but is not limited to the higher valuing of preprints and other forms of early sharing of ideas to enhance transparency and public discussion. We added to the third point:

The scientific community should be encouraged to take advantage of such platforms or other outlets for early sharing and discussion of their ideas and results.

3. Line 217: "the" should be "they".

Author response: The word 'the' was replaced by 'they'.

4. Line 265: "could" should be "could".

Author response: The word 'could' was corrected.

5. The sentence from 618 to 620 needs to be improved for better clarity.

**Author response:** The original sentence "Initially, the publication costs of ANGEO were largely covered by institutional subscriptions, before it was converted into an OA journal in 2009, so that authors had to pay the APCs in most cases." was changed to

Initially, the publication costs of ANGEO were largely covered by institutional subscriptions. In 2009, ANGEO was converted into an OA journal in 2009. As of then, authors had to pay individual APCs, which in many cases may not have been covered by institutional agreements (Section 4.1.7).

6. Line 911, "Initially" is used in the preceding sentence. It can be removed in the second sentence.

**Author response:** The word 'initially' was removed in the second sentence.

7. Line 1022: "references", do you mean "referees"?

Author response: Yes, we meant 'referees' and corrected it accordingly.

8. Line 1139: "If data are not publicly accessible at the time of publication, the data statement must specify when and where they will become available and how readers can access them until then." Is there a follow-up to make sure that the authors indeed follow what they say?

**Author response:** Currently, there is no standard follow-up on such data statements. If readers are not successful in requesting data directly from the corresponding authors, the handling journal editor may be contacted by the authors who can then follow up individually.

\_\_\_\_\_

**Referee #2, Ludo Waltman**

This paper provides a detailed overview of the approach to open access publishing, preprinting and open peer review developed by the journal Atmospheric Chemistry and Physics, as well as other journals of the European Geosciences Union (EGU), in close collaboration with the publisher Copernicus.

I very much enjoyed reading the paper. It offers in-depth insights into the historical development of the open science practices of the EGU journals, the motivations for developing these practices, and the development of similar practices by other journals and publishers. The paper also provides detailed statistics demonstrating the level of adoption of the open science practices of the EGU journals.

Additionally, the paper shows that EGU has been an absolute frontrunner in this space. Nowadays there are many more journals and platforms adopting open science practices similar to those used by the EGU journals, and some of these journals and platforms are regularly praised for their contribution to innovation in scientific publishing. However, it needs to be acknowledged that many of these innovations were already in use by the EGU journals two decades ago.

The authors also make a strong case for open peer review: "Such disclosure of reviewer comments upon

publication of final papers should be regarded a minimum standard for OA publishing, in order to counteract low scientific standards of (semi-)predatory and fraudulent journals that are solely motivated by the publishers' financial interests. The mere post-publication of reviewer reports, however, inherently leads to a kind of bias and loss of information because only the reports of finally accepted papers are shown, whereas the reviews for rejected manuscripts are lost." (l. 259-263). I fully agree with this argument, and I fully support the authors' recommendation to "introduce appropriate elements of transparency and open peer review in all peer-reviewed OA publications" (l.1213).

**Specific comments**

- 1. 246, "concerns about the rigor of peer review by the Web of Science": Web of Science evaluates journals to decide whether they can be selected for indexing in its databases. I think it is confusing to refer to this process as 'peer review'. The process is quite different from what is usually understood as peer review. Author response: We thank the referee for pointing out this unclear text. We did not mean to suggest that peer review is being conducted by the Web of Science. We rephrased the sentence as follows: However, this concept has led to major controversy among the *eLife* editors (Else, 2022; Abbott, 2023) and to concerns about the rigor of peer review and indexing by the Web of Science (Brainard, 2024a,b; eLife, 2024; Stern, 2024; Barbour et al., 2025; eLife, 2025).
- l. 272, "referee reports can be entered to platforms like ORCID or Publons": Is this correct? It seems that Publons doesn't exist anymore. It has partly been integrated into the Web of Science platform. **Author response:** The referee is correct. In 2022, Publons has been integrated into the Web of Science. As of then, peer reviewer reports are tracked by the Web of Science Reviewer Recognition Feature. We acknowledge this now in the text:

In addition, referee reports can be entered to platforms like ORCID or Publons the Web of Science Reviewer Recognition tool (formerly 'Publons' an independent platform (2012 - 2017, acquired by Clarivate in 2017, providing additional recognition for these scientific contributions.

l. 347, "whereas the former was fully sponsored by research funders, institutions and societies": I don't think this is correct. F1000Research applies an APC model.

**Author response:** The referee is right. While research funders, institutions and societies expressed their support for the new publishing model by F1000 in 2012, the platform was indeed fully financed via APCs. Accordingly, we changed the text as follows:

F1000 and PeerJ initially differed in their business model; whereas the former was fully financed through article processing charges (Lawrence, 2012) sponsored by research funders, institutions and societies, and the latter initially applied a membership-based model for authors which was extended in 2016 to allow for payments of individual articles.

- l. 362-364, "For example, the French-led Peer Community in (PCI, 2016) facilitates open peer review of preprints deposited on the Episciences platform (CCSD, 2017) or in the French 'Hyper Article en Ligne' repository (HAL, 2001).": My understanding is that PCI accepts preprints from a large number of preprint servers, not only the ones mentioned in this sentence.
- l. 366-367, "However, the small number of papers in the Peer Community Journals ( $\sim$ 100 per year, across a wide range of scientific disciplines) exemplifies its limited success and popularity.": I think this sentence is inaccurate. Only a subset of the articles reviewed and recommended by PCI are published in the Peer Community Journal. What matters is not the number of articles published in the Peer Community Journal but the total number of articles reviewed and recommended by PCI.

**Author response:** We thank the referee for pointing out some details about the PCI platform. We respond to both comments together since we rephrased the full text about the PCI (new text in bold):

Non-profit initiatives triggered the creation of funder-independent platforms. For example, the French-led *Peer Community in* (PCI, 2016) facilitates open peer review of preprints deposited on the *Episciences* platform (CCSD, 2017) or, in the French 'Hyper Article en Ligne' repository (HAL, 2001) or on several other preprint servers (OSF preprints, PaleorXiv, EcoEvorxiv, AfriArxiv, SocArXiv, and bioRxiv) (Peer Community In, 2013). Preprints posted on these servers can then be linked to one of 19 thematic PCIs for open peer review.

Upon acceptance of the paper by an editor, a paper can be either published at no-cost in the Peer Community Journal (PCI, 2016), or transferred to a 'PCI friendly journal' for potential publication, possibly without further peer review. In 2017, about 50 preprints were submitted and also recommended; these numbers increased to 518 and 240, respectively, in 2024, with each preprint receiving 2 - 3 reviews on average. In total, about 1800 preprints were linked to the PCI platform, 830 papers were recommended for

publication in either the Peer Community Journal or in an PCI friendly journal (aboout 50% each) (PCI Facts & Figures, 2024).

However, the small number of papers in the Peer Community Journals (~100 per year, across a wide range of scientific disciplines) exemplifies its limited success and popularity.

l. 903-905, "The APCs in Table 3 are comparable to fees applied on publishing platforms, such as those managed by F1000 (F1000 APCs; Open Research Europe (2020), Section 2.3.2). However, on these platforms, no type-setting and copy-editing is applied unlike for final articles in EGU journals": I am not sure if this is correct. It seems to me that F1000 does perform type-setting.

Author response: The referee is right that there is indeed typesetting performed on the F1000 platform (https://f1000research.com/for-authors/article-guidelines-the-production-process. We were misled by the APC break-down on the F100 website that does not specify the contribution of paper production to the total APCs (F1000 APCs) as the categories listed there, based on the Plan S Journal Comparison service that is being continued as of April 2025, do not exactly correspond to those as recommended by the FOAA (as applied by Copernicus Publications, Figure 11). We removed the sentence.

**Technical corrections**

l. 1022: I think 'references' should be 'referees'.

**Author response:** Yes, we meant 'referees' and corrected it accordingly.

\_\_\_\_\_

**Community Comment by Mingjin Tang**

One aspect that this manuscript has not emphasized (please accept my apologies if I miss it) is education and training the multi-stage open peer review provides, although this may not be one of original goals of the multi-stage open peer review.

From what I experience and observe, peer-review can be a mysterious and frightening process for early-career scientists. The multi-stage open peer review, provided by ACP and other journals, offers a unique and very helpful window for early career scientists to learn how to reply to reviews and how to provide reviews. I have benefitted a lot from it, and from time to time tell my students/postdoc that this is a very good way to learn how to provide/reply to reviews.

**Author response:** Indeed, the full record of the preprints in their original and revised forms, accompanied by the author responses and referee, community and editor comments are excellent material for teaching and training. We make this aspect more explicit throughout the manuscript:

p. 2 (Introduction), additional bullet point: - Educational value of public access to scientific communication and discussion, enabling everyone to follow and learn from real examples of how scientific critiques are addressed and how consensus can be reached, or how disagreement can be handled in a rational and constructive way.

Section 2.2 In addition, deleting the discourse on papers that do not ultimately result in journal publication diminishes the educational value of open peer review, as these examples also provide important learning opportunities and orientation for all involved parties.

Section 4.1.5 (new text in bold): To broaden the pool of referees and to particularly increase the number of ECSs as peer reviewers, EGU organized hands-on peer review trainings in 2023 and 2024 for its members, drawing on the extensive collection of preprints and referee reports as practical training resources.

**References**

Abbott, A.: Strife at eLife: inside a journal's quest to upend science publishing, Nature, 615, 780–781, https://doi.org/10.1038/d41586-023-00831-6, 2023.

Barbour, G., Carter, C., Coates, J., Cobey, K. D., Corker, K. S., Gadd, E., Kramer, B., Lawrence, R., Méndez, E., Neylon, C., Pölönen, J., Stern, B., and Waltman, L.: Criteria for Bibliographic Databases in a Well-Functioning Scholarly Communication and Research Assessment Ecosystem, URL https://upstream.force11.org/criteria-for-bibliographic-databases/, 2025.

- Brainard, J.: Open-access journal elife will lose its 'impact factor' over controversial publishing model, Science, https://doi.org/10.1126/science.zycyo78, 2024a.
- Brainard, J.: Web of Science index puts eLife 'on hold' because of its radical publishing model, Science, Scienceins, https://doi.org/10.1126/science.zjs3ept, 2024b.
- CCSD: Centre pour la Communicaiton Scientifique Directe, Episciences, open access overlay journals, URL https://www.episciences.org/, 2017, last access: 12 June 2025.
- eLife: Changes to eLife's indexing status in Web of Science and Scopus, URL https://elifesciences.org/inside-elife/ae620829/, 2024.
- eLife: The eLife Model: Two-year update, URL https://elifesciences.org/inside-elife/8947f033/the-elife-model-two-year-update, 2025, last access: 15 Apr 2025.
- Else, H.: eLife won't reject papers once they are under review what researchers think, Nature, https://doi.org/10.1038/d41586-022-03534-6, 2022.
- F1000 APCs: URL https://f1000research.com/for-authors/article-processing-charges, last access: 12 June 2025.
- HAL: Hyper Articles en Ligne, Center for Direct Scientific Communication (CCSD), URL https://about.ha l.science/en/, 2001, last access: 12 June 2025.
- Lawrence, R.: Publishing fees released, URL https://blog.f1000.com/2012/09/20/publishing-fees-released/, 2012, last access: 12 June 2025.
- PCI: Peer Community in, URL https://peercommunityin.org/, 2016, last access: 12 June 2025.
- PCI Facts & Figures: Partners and Supporters, URL https://peercommunityin.org/current-pcis/, 2024, last access: 12 June 2025.
- Peer Community In: Partners and Supporters, URL https://peercommunityin.org/pci-network/, 2013, last access: 12 June 2025.
- Stern, B.: How the Web of Science takes a step back, cOAlitionS Blog, URL https://www.coalition-s.org/blog/how-the-web-of-science-takes-a-step-back/, 2024, last access: 12 June 2025.